 

# Insulin regulates POMC neuronal plasticity to control glucose metabolism

Garron T Dodd[1,2†], Natalie J Michael[1,3†], Robert S Lee-Young[1,2,4], Salvatore P Mangiafico[5], Jack T Pryor[3,6], Astrid C Munder[1,3], Stephanie E Simonds[1,3], Jens Claus Brüning[7,8,9,10,11], Zhong-Yin Zhang[12], Michael A Cowley[1,3], Sofianos Andrikopoulos[5], Tamas L Horvath[13,14], David Spanswick[1,3,6]*, Tony Tiganis[1,2,4]*

[1]Metabolism, Diabetes and Obesity Program, Monash Biomedicine Discovery Institute, Monash University, Melbourne, Australia; [2]Department of Biochemistry and Molecular Biology, Monash University, Victoria, Australia; [3]Department of Physiology, Monash University, Victoria, Australia; [4]Monash Metabolic Phenotyping Facility, Monash University, Victoria, Australia; [5]Department of Medicine (Austin Hospital), The University of Melbourne, Melbourne, Australia; [6]Warwick Medical School, University of Warwick, Coventry, United Kingdom; [7]Department of Neuronal Control of Metabolism, Max Planck Institute for Metabolism Research, Cologne, Germany; [8]Center for Endocrinology, Diabetes, and Preventive Medicine, University Hospital Cologne, Cologne, Germany; [9]Excellence Cluster on Cellular Stress Responses in Aging-Associated Diseases, University of Cologne, Cologne, Germany; [10]Center for Molecular Medicine Cologne, University of Cologne, Cologne, Germany; [11]National Center for Diabetes Research, Neuherberg, Germany; [12]Department of Medicinal Chemistry and Molecular Pharmacology, Purdue University, West Lafayette, United States; [13]Program in Integrative Cell Signaling and Neurobiology of Metabolism, Department of Comparative Medicine, Yale University School of Medicine, New Haven, United States; [14]Department of Anatomy and Histology, University of Veterinary Medicine, Hungary, Europe

**\*For correspondence:**
David.Spanswick@monash.edu (DS);
Tony.Tiganis@monash.edu (TT)

[†]These authors contributed equally to this work

**Competing interests:** The authors declare that no competing interests exist.

**Abstract** Hypothalamic neurons respond to nutritional cues by altering gene expression and neuronal excitability. The mechanisms that control such adaptive processes remain unclear. Here we define populations of POMC neurons in mice that are activated or inhibited by insulin and thereby repress or inhibit hepatic glucose production (HGP). The proportion of POMC neurons activated by insulin was dependent on the regulation of insulin receptor signaling by the phosphatase TCPTP, which is increased by fasting, degraded after feeding and elevated in diet-induced obesity. TCPTP-deficiency enhanced insulin signaling and the proportion of POMC neurons activated by insulin to repress HGP. Elevated TCPTP in POMC neurons in obesity and/or after fasting repressed insulin signaling, the activation of POMC neurons by insulin and the insulin-induced and POMC-mediated repression of HGP. Our findings define a molecular mechanism for integrating POMC neural responses with feeding to control glucose metabolism.
DOI: https://doi.org/10.7554/eLife.38704.001

## Introduction

Insulin acts on peripheral tissues including liver, muscle and adipose tissue to directly control glucose metabolism, while also acting in the brain to concordantly regulate nutrient fluxes, feeding behaviour and energy homeostasis (*Varela and Horvath, 2012*). The CNS effects of insulin on metabolism are

mediated by different brain regions, in particular the hypothalamus, acting via autonomic circuits to influence peripheral organs, including the pancreas, liver, white adipose tissue and brown adipose tissue to modulate insulin secretion, endogenous glucose production and glucose uptake (*Dodd et al., 2015*; *Könner et al., 2007*; *Obici et al., 2002b*; *Pocai et al., 2005*; *Steculorum et al., 2016*; *Vogt and Brüning, 2013*; *Vogt et al., 2014*). Although our understanding of the neural circuitry controlling feeding behaviour and energy expenditure has grown considerably in the last few years, the neural processes by which insulin elicits its effects on glucose metabolism are less clear.

The insulin receptor (IR) is widely expressed throughout the brain (*Havrankova et al., 1978*). Mice lacking the IR in the brain exhibit whole-body insulin resistance accompanied by increased food intake and obesity (*Brüning et al., 2000*). The contributions of the IR in controlling feeding and energy expenditure are best understood in the arcuate nucleus (ARC) of the hypothalamus, a specialized brain region comprising of neurons proximal to the fenestrated capillaries of the median eminence. This position allows ARC neurons to respond to peripheral circulating factors such as leptin and insulin that convey information on the nutritional and metabolic status of the organism. In particular, the ARC comprises of two opposing neuronal populations, the appetite suppressing proopiomelanocortin (POMC)-expressing neurons and the orexigenic agouti-related peptide (AgRP)-expressing neurons that co-express neuropeptide Y (NPY) and γ-aminobutyric acid (GABA). The activation of POMC neurons results in POMC processing to generate α-melanocyte stimulating hormone (α-MSH), which is released and agonises melanocortin-4 receptors (MC4R) on neurons in other regions of the brain, such as the paraventricular nucleus of the hypothalamus (PVH) (*Varela and Horvath, 2012*). The activation of AgRP/NPY neurons promotes the synthesis and release of AgRP and GABA that antagonise α-MSH/MC4R interactions and directly inhibit POMC neurons, respectively (*Atasoy et al., 2012*; *Cowley et al., 2001*; *Cowley et al., 2003*; *Tong et al., 2008*).

Insulin binds to its tyrosine kinase receptor and signals via the phosphatidylinositol 3-kinase (PI3K)/protein kinase PKB/AKT pathway to hyperpolarise and inhibit AgRP/NPY neurons and repress *Agrp* expression (*Könner et al., 2007*; *Spanswick et al., 2000*; *Varela and Horvath, 2012*; *Vogt and Brüning, 2013*; *Zhang et al., 2015*). Leptin also hyperpolarizes and inhibits NPY/AgRP neurons and represses *Npy/Agrp* expression acting via several pathways, including the Janus-activated kinase (JAK)−2/signal transducer and activator of transcription (STAT)−3 pathway and the PI3K/AKT pathway (*Cowley et al., 2001*; *Elias et al., 1999*; *van den Top et al., 2004*; *Varela and Horvath, 2012*; *Zhang et al., 2015*). In this way leptin and insulin act to alleviate the inhibitory constraints on POMC neurons and the melanocortin response. IR signaling in AgRP neurons is important for the control of peripheral glucose metabolism (*Dodd et al., 2018*; *Könner et al., 2007*) acting through vagal efferents to the liver and α7-nicotinic acetylcholine receptors to repress hepatic glucose production (HGP) (*Kimura et al., 2016*). Central insulin signaling suppresses HGP by activating Kupffer cells to release interleukin- 6 (IL-6) (*Obici et al., 2002a*; *Obici et al., 2002b*; *Pocai et al., 2005*). IL-6 in turn acts on hepatocytes via STAT-3 to repress the expression of gluconeogenic enzymes such as glucose-6-phosphatase (encoded by *G6pc*) (*Inoue et al., 2006*; *Inoue et al., 2004*). IR deletion in AgRP neurons in mice results in defective inhibition of HGP accompanied by decreased *Il6* and increased *G6pc* hepatic gene expression (*Könner et al., 2007*).

Leptin depolarises and activates POMC neurons and promotes *Pomc* expression and α-MSH secretion to repress feeding, increase energy expenditure and repress HGP (*Berglund et al., 2012*; *Caron et al., 2018*; *Varela and Horvath, 2012*). By contrast insulin has been traditionally viewed as an inhibitor of POMC neuronal excitation, hyperpolarising POMC neurons through the engagement of ATP-sensitive K$^+$ channels (*Hill et al., 2008*; *Könner et al., 2007*; *Plum et al., 2006*; *Spanswick et al., 2000*; *Williams et al., 2010*). More recent studies have shown that insulin can depolarise and activate POMC neurons via the activation of transient receptor potential (TRPC)−5 channels (*Qiu et al., 2018*; *Qiu et al., 2014*), whereas several studies have shown that insulin can promote *Pomc* expression and the melanocortin response (*Varela and Horvath, 2012*). The advent of single-cell transcriptomics has yielded unprecedented insight into the heterogeneity of hypothalamic cell types and has led to the identification of different subsets of POMC- and AgRP-expressing neurons that exhibit distinct transcriptional responses to changing energy status (*Campbell et al., 2017*; *Chen et al., 2017*; *Henry et al., 2015*; *Lam et al., 2017*). Such heterogeneity may at least in part explain the seeming discordant effects of insulin on POMC neuronal excitability, but this remains to be formally examined. Irrespective, whereas insulin signaling in AgRP neurons overtly

affects glucose metabolism through the control of HGP and brown adipose tissue glucose uptake (*Könner et al., 2007*; *Shin et al., 2017*), gene deletion and pharmacogenetic experiments have suggested that insulin signaling in POMC neurons may not be so important; neither IR deletion in POMC neurons (*Könner et al., 2007*; *Shin et al., 2017*) nor the acute activation of POMC neurons using DREADDs (*Steculorum et al., 2016*) influences glucose metabolism. Nonetheless, the combined deletion of the IR plus leptin receptor in POMC neurons profoundly affects glucose homeostasis and results in systemic insulin resistance and increased HGP (*Hill et al., 2010*). Thus, the contribution of insulin signaling in POMC neurons on glucose metabolism and energy expenditure may be more nuanced and dictated by the hormonal milieu.

We report that insulin can elicit discordant effects on the electrophysiological activity of POMC neurons and that this is regulated by T-cell protein tyrosine phosphatase (TCPTP) (*Tiganis, 2013*), an IR phosphatase whose abundance in the ARC is increased by fasting and repressed by feeding (*Dodd et al., 2017*). Moreover we demonstrate that the regulation of IR signaling by TCPTP dictates whether POMC neurons are activated or inhibited by insulin and that this serves to coordinate hepatic glucose metabolism with feeding and fasting. Our studies define a mechanism for linking POMC neuronal plasticity with the nutritional and metabolic state of the organism.

## Results

### TCPTP defines POMC neurons that are activated or inhibited by insulin

We explored the insulin-induced activation of GFP-positive ARC POMC neurons in coronal slices from *ad libitum* fed *Pomc*-eGFP reporter mice using the whole-cell patch clamp technique; previous studies have confirmed that the GFP reporter faithfully marks POMC neurons in adult mice (*Padilla et al., 2010*). We monitored for changes in membrane potential and firing frequency in response to 100 nM recombinant insulin and recorded from 53 rostral or central/caudal POMC neurons that had an average resting membrane potential of $-42.3 \pm 2.1$ mV. The majority of POMC neurons were either inhibited by insulin (27/53, 50.9%) with a significant membrane hyperpolarisation (baseline $-42.5 \pm 1.2$ mV; insulin $-45.9 \pm 1.8$ mV) and decreased firing rate (baseline $1.09 \pm 0.20$ Hz; insulin $0.2 \pm 0.06$ Hz) or were unaffected by insulin (20/53, 37.7%) with minimal changes in membrane potential (baseline $-46.8 \pm 1.7$ mV; insulin $-45.7 \pm 1.8$ mV) and non-significant changes in firing rate (baseline $1.32 \pm 0.51$ Hz; insulin $1.08 \pm 0.41$ Hz, *Figure 1a–c*). A smaller subset of POMC neurons (6/53, 11.3%) were activated by insulin exhibiting significant membrane depolarisation (baseline $-45.9 \pm 2.4$ mV; insulin $-40.1 \pm 2.5$ mV) and increased firing rates (baseline $0.99 \pm 0.37$ Hz; insulin $1.69 \pm 0.60$ Hz, *Figure 1a–c*). Although the unresponsive and inhibited POMC neurons did not exhibit any distinct anatomical distribution, the excited POMC neurons were located more medially and comparatively closer to the third ventricle. (*Figure 1d*). All subsets of insulin-responsive POMC neurons were found across the rostral to caudal extent of the ARC.

We have reported previously that POMC neurons in the ARC differentially express the tyrosine phosphatase TCPTP (*Dodd et al., 2015*) and that hypothalamic TCPTP expression, including that in AgRP and POMC neurons, is induced by fasting and repressed in fed or satiated mice where TCPTP protein is additionally degraded and eliminated (*Dodd et al., 2017*). To determine the extent to which the differential insulin responses in POMC neurons from *ad libitum* fed (at 9–11 am) mice may relate to the expression of TCPTP, we intracellularly labelled recorded cells with biocytin and processed the coronal slices for TCPTP immunohistochemistry. We found that the majority of inhibited (25/27) and around half of all unresponsive (11/20) POMC neurons in the patch-clamped coronal slices from *ad libitum* fed *Pomc*-eGFP mice expressed TCPTP (*Figure 1e–f*). On the other hand, most (5/6) of POMC neurons that were excited by insulin did not express TCPTP (*Figure 1f*). Therefore, these results are consistent with TCPTP status determining whether POMC neurons are excited by insulin, or whether they are inhibited by insulin. The extent to which TCPTP may influence non-responsive neurons is unclear and it possible that these neurons do not express the IR. Nonetheless, to explore the influence of TCPTP status on the excitation versus inhibition of POMC neurons we crossed mice in which TCPTP (encoded by *Ptpn2*) had been deleted in POMC neurons (*Pomc*-Cre; *Ptpn2*^*fl/fl*: POMC-TC) on the Z/EG reporter background (POMC-TC;Z/EG) (*Novak et al., 2000*), where Cre-mediated recombination can be tracked by the expression of GFP. TCPTP deleted GFP-positive POMC neurons in coronal slices from *ad libitum* fed adult mice were subjected to whole-cell



**Figure 1.** POMC neurons display differential responses to insulin dependent on TCPTP. Whole-cell patch clamp recording of hypothalamic POMC neurons in response to insulin (100 nM) in *Pomc*-eGFP mice. (**a**) Representative traces of individual POMC neurons displaying either excitation (depolarisation) or inhibition (hyperpolarisation) or POMC neurons that are non-responsive to insulin. (**b**) Grouped POMC population insulin responses, (**c**) membrane potential change and firing frequencies recorded from the entire rostral-caudal extent of the hypothalamic POMC neuronal population.
*Figure 1 continued on next page*

*Figure 1 continued*

Electrophysiological responses were measured in n = 53 independent neurons across 8–10 mice and analysed using a two-tailed t-test. (**d**) Camera lucida image depicting topographical localisation of insulin-responsive POMC neuronal subtypes. During recordings, patch clamped POMC neurons were filled with biocytin and post-recording ex vivo sections were incubated in paraformalydhyde and processed for biocytin, GFP and TCPTP immunohistochemistry. (**e**) Representative micrographs depicting TCPTP positive and negative patch-clamped POMC neurons and (**f**) insulin responsiveness and TCPTP expression correlation. To determine the functional role of TCPTP in POMC neuronal insulin sensitivity, whole-cell patch clamp recording of POMC neurons in response to insulin (100 nM) were performed in (**g-h**) *Pomc*-Cre;*Ptpn2*$^{fl/fl}$ (POMC-TC) mice on the on the Z/EG reporter background or (**i-j**) in *Pomc*-eGFP mice pre-treated with vehicle or TCPTP inhibitor (compound 8, 20 nM). (**g, i**) Grouped POMC population insulin responses, (**h, j**) membrane potential change and firing frequencies recorded from the entire rostral-caudal extent of the hypothalamic POMC neuronal population. (**k–l**) Whole-cell patch clamp recording of POMC neurons from 8 to 10 week high fat fed *Pomc*-eGFP mice pre-treated with vehicle or TCPTP inhibitor (compound 8, 20 nM) in response to insulin (100 nM). (**k**) Grouped POMC population insulin responses, (**l**) membrane potential change and firing frequencies recorded from the entire rostral-caudal extent of the hypothalamic POMC neuronal population. Results shown are means ±SEM for the indicated number of cells in pie charts. Electrophysiological responses were measured in (**g-h**) 33, (**i–j**) 58 and **k-l**) 34 (vehicle) and 35 (TCPTP inhibitor) independent neurons across 8–10 mice and analysed using a two-tailed t-test. (**m**) 8–10 week-old POMC-TC or *Ptpn2*$^{fl/fl}$ overnight fasted male mice were administered (intraperitoneal) saline or 0.85 mU/g insulin and 90 min later brains fixed with paraformaldehyde and processed for paraventricular hypothalamus (PVH) c-Fos immunoreactivity. Data was analysed using a two-way ANOVA followed by Tukey multiple comparison test. Representative images and quantified (means ± SEM) results are shown for the indicated number of cells/mice. (**b, g, l, k**) date rounded up to the nearest integer.

DOI: https://doi.org/10.7554/eLife.38704.002

The following figure supplements are available for figure 1:

**Figure supplement 1.** Diet-induced obesity promotes TCPTP expression in POMC neurons.
DOI: https://doi.org/10.7554/eLife.38704.003

**Figure supplement 2.** Intraperitoneal insulin-induced hypoglycaemia is not altered in POMC-TC mice.
DOI: https://doi.org/10.7554/eLife.38704.004

recordings, monitoring once more for changes in membrane potential and firing frequency in response to insulin. Recordings were undertaken on 33 ARC POMC neurons with a resting membrane potential of −40.1 ± 1.2 mV. Strikingly TCPTP-deficiency significantly increased the number of cells that were excited (12/33, 36.4%) in response to insulin and decreased the number of inhibited (12/33, 36.4%) POMC neurons; TCPTP-deficiency tended to decrease the number of unresponsive (9/33, 27.3%) POMC neurons but this was not significant (*Figure 1g–h*; *Supplementary file 1*). Excited neurons were significantly depolarised in response to insulin (baseline membrane potential −42.1 ± 1.8 mV; insulin −37.9 ± 1.7 mV) and firing rate increased (baseline 0.69 ± 0.33 Hz; insulin 1.58 ± 0.66 Hz), whereas insulin-inhibited neurons were significantly hyperpolarised (baseline membrane potential −43.0 ± 1.8 mV; insulin −46.8 ± 2.3 mV) and the firing rate reduced (baseline 0.63 ± 0.25 Hz; insulin 0.10 ± 0.05 Hz) in response to insulin (*Figure 1g–h*). Given that the *Pomc*-Cre transgene can delete in roughly 25% of NPY/AgRP neurons in early development (*Padilla et al., 2010*; *Xu et al., 2018*), we cannot exclude that a proportion of the inhibited neurons were in fact AgRP neurons that are inhibited by insulin (*Könner et al., 2007*; *Spanswick et al., 2000*). Nonetheless these results are consistent with TCPTP-deficiency, which emulates the fed state (*Dodd et al., 2017*), increasing the proportion of POMC neurons activated and decreasing the proportion of POMC neurons inhibited by insulin.

To independently assess the impact of TCPTP deficiency on the excitation of POMC neurons in response to insulin we also recorded GFP-positive ARC POMC neurons from *ad libitum* (at 9–11 am) fed *Pomc*-eGFP reporter mice that had been treated ICV with the highly potent and selective TCPTP inhibitor compound 8 (*Zhang et al., 2009*) that we have shown previously elicits effects in wild-type but not TCPTP-deficient mice (*Loh et al., 2011*). Strikingly, TCPTP inhibition significantly altered POMC responses to insulin such that the proportion of GFP-positive POMC neurons activated by insulin increased to 33% (19/58 cells) (*Figure 1i*; *Supplementary file 2*). These results indicate that the deletion or inhibition of TCPTP can increase the proportion of POMC neurons that are activated by insulin.

Next, we determined whether the converse may be true and if increased TCPTP might be associated with an increased proportion of POMC neurons that are inhibited by insulin. To this end we recorded from GFP-positive POMC neurons in coronal slices from *Pomc*-eGFP mice that had been fed a high-fat diet (23% fat) for 12 weeks to render them obese. We have reported previously that hypothalamic TCPTP levels are elevated in diet-induced obesity (*Loh et al., 2011*), whereas more

recently we have shown that feeding-induced repression of hypothalamic TCPTP expression is defective in obesity resulting in elevated TCPTP (*Dodd et al., 2017*). Consistent with this, TCPTP protein levels, as assessed by immunoblotting mediobasal hypothalamic homogenates from fed [allowed to feed for 4 hr after start of dark cycle till satiated (*Dodd et al., 2017*)] mice that had been administered a high fat diet for 12 weeks, were increased by $6.3 \pm 0.49$ fold when compared to those in corresponding chow-fed lean controls (*Figure 1—figure supplement 1a–b*). Importantly, the increase in TCPTP occurred in hypothalamic neurons of the ARC, including POMC neurons, as assessed by the increased coincidence of TCPTP with GFP-positive POMC neurons (*Figure 1—figure supplement 1c*) and was accompanied by the repression of insulin-induced PI3K/AKT signaling that was restored by TCPTP-deficiency (*Figure 1—figure supplement 1d*). Accordingly, we reasoned that sustained/elevated hypothalamic TCPTP levels in diet-induced obesity might shift POMC neural responses so that the majority of POMC neurons are unresponsive or inhibited by insulin. Consistent with this, we found that in high-fat-fed obese mice the proportion of GFP-positive POMC neurons inhibited by insulin ex vivo increased from 51% (27/53 cells) to 65% (22/34 cells; p=0.045; *Supplementary file 2*), whereas the proportion of cells being activated fell from 11% (6/53 cells) to 3% (1/34 cells; p=0.027, *Figure 1k*; *Supplementary file 2*). Insulin-inhibited neurons were significantly hyperpolarised (baseline membrane potential $-45.8 \pm 0.9$ mV; insulin $-49.9 \pm 1.3$ mV) and the firing rate reduced (baseline $1.33 \pm 0.37$ Hz; insulin $0.11 \pm 0.05$ Hz) in response to insulin (*Figure 1l*). To determine the extent to which this might be attributable to the increase in TCPTP in POMC neurons we recorded GFP-positive neurons in slices prepared from 12 week high-fat-fed mice that had been treated ICV with compound 8 (*Figure 1k*). Strikingly, TCPTP inhibition significantly altered POMC responses to insulin such that only 20% (7/35 cells) of GFP-positive POMC neurons were inhibited by insulin, whereas the proportion of cells being activated increased from 3% (1/34 cells) to 37% (13/35; p<0.001, *Figure 1k–l*; *Supplementary file 2*). Inhibited neurons were hyperpolarised (baseline membrane potential $-47.6 \pm 1.3$ mV; insulin $-49.9 \pm 1.6$ mV) and the firing rate reduced in the presence of insulin (baseline $1.20 \pm 0.78$ Hz; insulin $0.07 \pm 0.05$ Hz), whereas activated neurons were significantly depolarised (baseline membrane potential $-46.0 \pm 0.8$ mV; insulin $-42.6 \pm 0.7$ mV) and the firing rate increased (baseline $0.53 \pm 0.22$ Hz; insulin $0.83 \pm 0.30$ Hz) in response to insulin (*Figure 1l*). Therefore, these results are consistent with increased/sustained hypothalamic TCPTP in obesity orchestrating a switch in neuronal excitability such that POMC neurons are principally inhibited by insulin.

Our results indicate that TCPTP status may define POMC neural responses to insulin so that feeding-associated diurnal fluctuations in TCPTP or elevated TCPTP levels in obesity might dictate whether POMC neurons are activated by insulin or are otherwise inhibited or remain unresponsive. To determine whether TCPTP status and altered POMC neuronal responses to insulin might be of functional relevance in vivo we monitored the influence of TCPTP deficiency in POMC neurons on the activation of second order neurons in the PVH in mice that had been fasted and administered a bolus of insulin. TCPTP-deficiency in POMC-TC mice increased c-Fos staining (a surrogate marker of neuronal activation) in the PVH of fasted mice and this was exacerbated by insulin (*Figure 1m*); although systemic insulin administration resulted in hypoglycemia, this was similar in *Ptpn2*<sup>fl/fl</sup> and POMC-TC mice (*Figure 1—figure supplement 2*). These results are consistent with TCPTP deficiency promoting the activation of POMC neurons in vivo. Taken together these results establish the potential for POMC neurons in the ARC to be either activated or inhibited by insulin. Moreover, our studies indicate that response of POMC neurons to insulin may be dictated by the levels of TCPTP, so that diurnal fluctuations in TCPTP in ARC POMC neurons [TCPTP is increased in the fasted state and eliminated in the fed/satiated state (*Dodd et al., 2017*)] may alter the proportion of neurons that are activated or inhibited by insulin and thereby alter the functional melanocortin output response.

## The activation or inhibition of POMC neurons regulates hepatic glucose metabolism

Neurons in the hypothalamus and other regions of the brain can respond to peripheral signals such as insulin, leptin, glucose and free fatty acids, to influence peripheral glucose and lipid metabolism (*Berglund et al., 2012*; *Buettner et al., 2008*; *Claret et al., 2007*; *Gelling et al., 2006*; *Lam et al., 2005*; *Obici et al., 2002b*; *Parton et al., 2007*; *Shin et al., 2017*; *Varela and Horvath, 2012*). In particular, insulin-responsive ARC neurons can function via vagal efferent inputs to the liver to

repress HGP (*Kimura et al., 2016*; *Obici et al., 2002a*; *Obici et al., 2002b*; *Pocai et al., 2005*). Gene deletion experiments indicate that AgRP neurons are instrumental in the insulin-mediated repression of hepatic gluconeogenesis (*Könner et al., 2007*). On the other hand IR deletion in POMC neurons (*Könner et al., 2007*) or the acute pharmacogenetic activation of POMC neurons in fasted mice have no effect on glucose metabolism (*Steculorum et al., 2016*). Since our studies indicate that distinct POMC neuronal subsets exist that can be inhibited (hyperpolarised) or activated (depolarised) by insulin, we reasoned that previous gene deletion studies may have been confounded by the existence of POMC neurons with potentially diametrically opposing consequences on glucose metabolism. Accordingly, we compared the effects of chronically inhibiting versus activating POMC neurons on hepatic glucose metabolism. To this end we took advantage of the $G_i$-coupled hM4Di inhibitory DREADD (designer receptors exclusively activated by designer drugs), which is activated by clozapine-N-oxide (CNO) and induces neuronal silencing (*Alexander et al., 2009*; *Armbruster et al., 2007*; *Ferguson et al., 2011*), versus the $G_q$-coupled hM3Dq stimulatory DREADD, which depolarises neurons and induces firing in response to CNO. Recombinant adeno-associated viruses (rAAVs) capable of expressing either the hM4Di or the hM3Dq DREADDs fused to mCherry in a Cre-dependent manner (*Krashes et al., 2011*) were administered into the ARC of 12-week-old *Pomc*-Cre mice on the *Pomc*-eGFP reporter background; post-mortem analyses confirmed mCherry expression in 77.5 ± 12.0% and 74.1 ± 13.9% of GFP positive POMC neurons respectively (*Figure 2—figure supplement 1a–b*). mCherry was not expressed in non-GFP expressing neurons in the ARC (*Figure 2—figure supplement 1a–b*), consistent with the DREADDs being expressed specifically in POMC neurons. This is also consistent with studies showing that Cre-dependent AVV expression in the ARC of adult *Pomc*-Cre mice occurs specifically in POMC neurons (*Xu et al., 2018*). To assess the influence of POMC neuronal inhibition, we administered CNO in the dark phase, 4 hr after lights were turned off when mice were satiated (*Figure 2a–b*), so that HGP would be low and any effects otherwise arising from the POMC-mediated repression of feeding would be limited. This would allow us to examine if the insulin-mediated inhibition of POMC neurons might promote hepatic gluconeogensis. Mice expressing hM4Di-mCherry in POMC neurons were administered either a single dose of CNO 30 min before analysis, to examine the influence of acute POMC inhibition, or two doses of CNO over 6 hr to assess the effects of more sustained POMC neuronal inhibition and any accompanying effects on gene expression on hepatic glucose metabolism (*Figure 2a–c*). In the first instance, we measured HGP by performing pyruvate tolerance tests (pyruvate increases blood glucose by promoting gluconeogenesis). We found that the acute inhibition of POMC neurons had no effect, whereas the inhibition of POMC neurons over 6 hr increased blood glucose levels in response to pyruvate, consistent with the promotion of HGP (*Figure 2c*); CNO had no effect on the electrophysiological response of non-DREADD expressing POMC neurons (data not shown) and did not alter glucose metabolism (as assessed by measuring glucose excursions in response to pyruvate, glucose or insulin) in C57BL/6 control mice (*Figure 2—figure supplement 1c–e*). Hypothalamic neurons may influence HGP by regulating hepatic Il6 expression, which activates STAT3 in hepatocytes to suppress the expression of glucose-6-phosphatase (encoded by *G6pc*) and phosphoenolpyruvate carboxykinase (PEPCK; encoded by *Pck1*), rate-determining gluconeogenic enzymes (*Kimura et al., 2016*; *Obici et al., 2002a*; *Obici et al., 2002b*; *Pocai et al., 2005*). We found that the chronic, but not acute inhibition of POMC neurons was accompanied by the significant repression of hepatic *Il6* expression and diminished hepatic STAT3 Y705 phosphorylation (*Figure 2d–e*). This in turn was associated with increased expression of the gluconeogenic genes *Pck1* and *G6pc* (*Figure 2e*). These results are therefore consistent with POMC neuronal inhibition promoting hepatic gluconeogenesis. To explore this further we assessed the influence of chronic CNO administration on glucose turnover and hepatic and whole-body insulin sensitivity by subjecting mice to hyperinsulinemic euglycemic clamps (*Figure 2f–h*). We found that chronic POMC neuronal inhibition was associated with attenuated insulin-mediated repression of endogenous glucose production and an overall decrease in insulin sensitivity as assessed by the reduced glucose infusion rate (GIR) necessary to maintain euglycemia during the clamp and the reduced glucose disappearance rate (Rd; *Figure 2f–g*; *Figure 2—figure supplement 2a*). The attenuated repression of endogenous glucose production was accompanied by decreased *Il6* and increased hepatic *Pck1* and *G6pc* expression (*Figure 2h*), in keeping with the attenuated repression of HGP. Therefore, inhibition of POMC neurons promotes hepatic gluconeogenesis and glucose production and results in decreased whole-body insulin sensitivity.



**Figure 2.** Pharmacogenetic stimulation or inhibition of hypothalamic POMC neurons modulates hepatic glucose metabolism. 8-week-old *Pomc*-Cre mice (on the *Pomc*-eGFP reporter background) were bilaterally injected with rAAV-hSyn-DIO-hM4D(Gi)-mCherry into the ARC. (a) Inhibitory DREADD experimental paradigm schematic and (b) C57BL/6 feeding profile. (c–e) Two weeks post-AAV injection mice were administered vehicle or clozapine-N-oxide (CNO; 0.3 mg/kg, intraperitoneal) as indicated for either 30 min (one injection) or 6 hr (two injections 3 hr apart) prior to being processed for (c)

*Figure 2 continued on next page*

*Figure 2 continued*

pyruvate tolerance tests (0.75 mg/g; statistical differences were determined using a two-way ANOVA with repeated measures followed by Sidak multiple comparison test. In separate experiments mice were administered vehicle or CNO (0.3 mg/kg, intraperitoneal) over 6 hr (two injections 3 hr apart) and livers extracted for (**d**) immunoblotting (data was analysed using a two-tailed t-test) and (**e**) quantitative PCR, (data was analysed using a two-tailed t-test) or (**f–h**) mice were subjected to hyperinsulinemic-euglycemic conscious clamps (data was analysed using a two-way ANOVA with repeated measures followed by Sidak multiple comparison test). (**f**) Glucose infusion rates (GIR), (**g**) glucose disappearance rates (Rd) and basal and clamped endogenous glucose production (EGP; glucose appearance rate minus GIR) were determined and (**h**) livers extracted for quantitative PCR (data was analysed using a two-tailed t-test). (**i–l**) 8-week-old *Pomc*-Cre mice were bilaterally injected with rAAV-hSyn-DIO-hM3D(Gq)-mCherry into the ARC. (**i**) Excitatory DREADD experimental paradigm schematic. Two weeks post AAV injection mice were fasted overnight and administered vehicle or CNO (0.3 mg/kg, intraperitoneal) either 30 min (one injection) or 6 hr (two injections 3 hr apart) prior to being processed for (**j**) pyruvate tolerance tests (1.25 mg/g); significance was assessed using a two-way ANOVA with repeated measures followed by Sidak multiple comparison test. In a separate experiment, mice were fasted overnight and administered vehicle or CNO (0.3 mg/kg, intraperitoneal) as indicated over 6 hr (two injections 3 hr apart) and livers extracted for (**k**) immunoblotting and (**l**) quantitative PCR with significance being assessed using a two-tailed t-test in each case. Representative and quantified results are shown (means ± SEM) for the indicated number of mice.

DOI: https://doi.org/10.7554/eLife.38704.005

The following figure supplements are available for figure 2:

**Figure supplement 1.** DREADDs in ARC POMC neurons.

DOI: https://doi.org/10.7554/eLife.38704.006

**Figure supplement 2.** Clamped blood glucose levels in CNO-treated mice expressing inhibitory DREADDs in POMC neurons.

DOI: https://doi.org/10.7554/eLife.38704.007

To explore the influence of chronic POMC activation on hepatic glucose metabolism, mice expressing hM3Dq-mCherry in POMC neurons were fasted overnight (14 hr) so that HGP would be elevated and then administered a single dose of CNO 30 min before analysis, or two doses of CNO over 6 hr before subjecting mice to pyruvate tolerance tests or assessing hepatic gene expression and STAT3 signaling (*Figure 2i–l*). Chronic but not acute POMC neuronal activation resulted in the suppression of glucose excursion in response to pyruvate (*Figure 2j*) in keeping with the suppression of hepatic gluconeogenesis and was accompanied by elevated hepatic *Il6* expression, increased STAT3 Y705 phosphorylation and the significant repression of *Pck1* and *G6pc* expression (*Figure 2k–l*). Therefore, the chronic activation of POMC neurons represses gluconeogenesis and improves hepatic glucose metabolism. Taken together, our findings indicate that shifts in the proportion of POMC neurons that are activated or inhibited by insulin may have direct consequences on peripheral glucose metabolism.

## TCPTP in POMC neurons regulates hepatic glucose metabolism

We next explored whether alterations in TCPTP and the proportion of POMC neurons that are activated by insulin influence hepatic glucose metabolism. To this end we first determined whether deleting TCPTP, which results in a greater number of neurons being activated by insulin, represses hepatic gluconeogenesis and glucose output. We subjected *Ptpn2*^fl/fl^ and POMC-TC mice that had been fasted overnight and administered intracerebroventricular (ICV) insulin, to pyruvate tolerance tests (*Figure 3a*); ICV insulin administration did not alter plasma insulin levels in control C57BL/6 mice (*Figure 3—figure supplement 1a–b*). At doses of insulin that elicited little to no effect in *Ptpn2*^fl/fl^ control mice, we found that TCPTP deletion in POMC neurons resulted in improved pyruvate tolerance, as assessed by the suppression of pyruvate-induced increases in blood glucose levels (*Figure 3b*; *Figure 3—figure supplement 1c*). The improved pyruvate tolerance was accompanied by increased ICV insulin-induced hypothalamic *Pomc* and hepatic *Il6* gene expression as well as increased hepatic p-STAT-3 (*Figure 3c–e*). The hepatic expression of the gluconeogenic genes *Pck1* and *G6pc* expression was elevated in overnight fasted POMC-TC mice (*Figure 3d*), probably as a compensatory mechanism to prevent hypoglycemia. Nonetheless the increased hepatic STAT3 signaling in ICV insulin-administered POMC-TC mice was associated with the increased repression of hepatic *Pck1* and *G6pc* expression (*Figure 3d*). Therefore TCPTP-deficiency in POMC neurons enhances the insulin-hypothalamic-mediated repression of hepatic gluconeogenesis.

Next we determined the extent to which POMC TCPTP-deficiency influences whole-body glucose metabolism. TCPTP-deficiency was accompanied by lower fasted (6 hr) blood glucose and plasma insulin levels, in keeping with overall improvements in glucose metabolism and insulin sensitivity



**Figure 3.** Deletion of TCPTP in POMC neurons enhances the insulin-induced and POMC-mediated repression of hepatic gluconeogenesis. (a) Experimental paradigm schematic. 8–10 week-old POMC-TC or *Ptpn2^fl/fl* overnight fasted male mice were administered intracerebroventricularly (ICV) saline or insulin (0.1 mU/animal, 5 injections over 5 hr) as indicated and subjected 1 hr later to either (b) pyruvate tolerance tests (1 mg/g; areas under curves were determined and statistical significance assessed using a two-way ANOVA followed by Sidak multiple comparison test) or hypothalami and livers were extracted for (c-d) quantitative PCR (significance assessed using a one-way ANOVA followed by Tukey multiple comparison test) and (e) immunoblotting (significance assessed using a two-way ANOVA followed by Sidak multiple comparison test). Representative and quantified results are shown (means ± SEM) for the indicated number of mice.

DOI: https://doi.org/10.7554/eLife.38704.008

The following figure supplement is available for figure 3:

**Figure supplement 1.** Effects of central insulin administration in POMC-TC mice on glucose metabolism.

DOI: https://doi.org/10.7554/eLife.38704.009

(*Figure 4a–b*). This was reflected in greater reductions in blood glucose levels in response to insulin and improved glucose clearances in response to a bolus of glucose (*Figure 4c–d*). Moreover, excursions in blood glucose in response to pyruvate were attenuated in keeping with reduced endogenous glucose production (*Figure 4e*). To further explore the influence of POMC TCPTP-deficiency on glucose metabolism we subjected mice to hyperinsulinemic euglycemic clamps. We found that the glucose infusion rate (GIR) necessary to maintain euglycemia was significantly enhanced in POMC-TC mice, consistent with enhanced whole-body insulin sensitivity (*Figure 4f*). Importantly we found that the glucose disappearance rate (Rd), which primarily measures muscle and adipose tissue glucose uptake, was unaltered by TCPTP-deficiency under the conditions tested, whereas the insulin-induced repression of endogenous glucose production, a measure of hepatic and renal glucose production, was significantly enhanced in POMC-TC mice (*Figure 4f*; *Figure 4—figure supplement 1a–b*). The repressed endogenous glucose production was accompanied by decreased expression of hepatic gluconeogenic genes (*Pck1* and *G6pc*, *Figure 4g*). In keeping with the overall



**Figure 4.** Deletion of TCPTP in POMC neurons improves whole-body glucose homeostasis and represses HGP. Fed and fasted (a) blood glucose and (b) plasma insulin from 12-week-old POMC-TC or *Ptpn2*[fl/fl] male mice; significance was assessed using a two-way ANOVA followed by Tukey multiple comparison test. (c) 10–12 week-old male POMC-TC or *Ptpn2*[fl/fl] mice were subjected to (c) insulin (0.5 mU/g), (d) glucose (2 mg/g) or (e) pyruvate tolerance tests (1 mg/g); areas under curves were determined and statistical significance assessed using a two-tailed t-test. (f–g) Hyperinsulinemic-euglycemic clamps in 8–10 week-old POMC-TC or *Ptpn2*[fl/fl] fasted (4 hr) unconscious mice. GIR, Rd and basal and clamped EGP were determined, and statistical significance assessed using a two-tailed t-test. (g) Livers were extracted from clamped mice and processed for (g) quantitative PCR and statistical significance assessed using a two-tailed t-test. (h) 10-week-old POMC-TC or *Ptpn2*[fl/fl] male mice were fasted 4 hr and injected with saline or insulin (0.3 mU/g, intraperitoneal) and livers extracted after 15 min for immunoblotting and significance determined using a two-way ANOVA followed by Sidak multiple comparison test. Representative and quantified results are shown (means ± SEM) for the indicated number of mice.
DOI: https://doi.org/10.7554/eLife.38704.010

The following figure supplement is available for figure 4:

**Figure supplement 1.** Glucose metabolism in POMC-TC mice.
DOI: https://doi.org/10.7554/eLife.38704.011

improvement in insulin sensitivity we found that insulin-induced IR Y1162/Y1163 tyrosine phosphory-lation and activation and downstream PI3K/AKT signaling in the liver (as monitored by AKT Ser-473 phosphorylation) were dramatically enhanced by POMC TCPTP deficiency (*Figure 4h*). By contrast insulin-induced PI3K/Akt signaling was not altered in skeletal muscle or white adipose tissue (*Figure 4—figure supplement 1c–d*). Therefore, these findings are consistent with POMC TCPTP deficiency influencing hepatic glucose metabolism in two ways. First by enhancing the CNS insulin-induced and POMC-mediated repression of STAT3 signaling and gluconeogenic gene expression in the liver, and second by selectively enhancing the peripheral response of the liver to the actions of insulin.

## TCPTP in POMC neurons regulates hepatic glucose metabolism in response to feeding and fasting

Our studies indicate that TCPTP expressing and non-expressing POMC neurons exist in the hypo-thalami of *ad libitum* fed mice and that the expression of TCPTP dictates whether such neurons are inhibited (hyperpolarised) or activated (depolarised) by insulin. Moreover, our studies indicate that the presence or absence of TCPTP in POMC neurons determines the gluconeogenic status and insulin responsiveness of the liver. We have reported recently that TCPTP levels in the ARC exhibit diurnal fluctuations that are linked to feeding, so that fasting increases TCPTP expression and feeding represses TCPTP expression and actively promotes its degradation (*Dodd et al., 2017*). Such fluctuations in TCPTP were seen in both AgRP and POMC neurons (*Dodd et al., 2017*). Accordingly, we reasoned that elevated TCPTP in POMC neurons during fasting would render POMC neurons inhibited by insulin, to antagonise the CNS insulin-induced repression of hepatic glucose metabolism, whereas decreased TCPTP in response to feeding, would render POMC neurons activated by insulin, to facilitate the CNS insulin-induced repression of hepatic glucose metabolism. In this way feeding-associated fluctuations in TCPTP in POMC neurons would coordinate POMC neuronal responses and HGP to maintain euglycemia in response to feeding and fasting. To test this, we first compared the effects of centrally administered insulin on hepatic glucose metabolism in C57BL/6 mice that had been fed for 4 hr from the start of the dark cycle (*Figure 5a–c*) when mice are satiated (*Figure 2b*), to the corresponding mice where food was withdrawn at the start of the dark cycle (*Figure 5d–f*). We found that in fed mice, ICV administered insulin repressed hepatic gluconeogenesis as assessed by the attenuated glucose excursions in response to pyruvate (*Figure 5b*) and the inhibition of gluconeogenic gene expression (*Pck1* and *G6pc*; *Figure 5c*). This was accompanied by the promotion of hepatic *Il6* expression (*Figure 5c*). By contrast, in food-restricted mice (*Figure 5d*), ICV administered insulin had no effect on pyruvate-induced glucose excursions (*Figure 5e*) or the expression of *Pck1*, *G6pc* and *Il6* (*Figure 5f*). These results are in keeping with the feeding-induced repression of TCPTP and the activation of POMC by insulin, repressing hepatic glucose metabolism. Next, we specifically explored the contributions of TCPTP in POMC neurons. To this end we ICV administered insulin to food-restricted *Ptpn2$^{fl/fl}$* or POMC-TC mice and assessed hepatic glucose metabolism and gluconeogenic gene expression. As in food-restricted C57BL/6 mice, ICV insulin had no effect on glucose excursions in response to pyruvate in food-restricted *Ptpn2$^{fl/fl}$* mice (*Figure 5g*), nor did it alter the expression of *Pck1*, *G6pc* or *Il6* (*Figure 5h*). By contrast glucose excursions in vehicle-treated POMC-TC mice were reduced relative to *Ptpn2$^{fl/fl}$* mice and reduced further in response to ICV insulin administration (*Figure 5g*). As noted in overnight fasted POMC-TC mice (*Figure 3d*), the hepatic expression of *Pck1* and *G6pc* was elevated in vehicle-treated food-restricted POMC-TC mice (*Figure 5h*). Nonetheless, ICV administered insulin dramatically reduced hepatic *Pck1* and *G6pc* expression below that in *Ptpn2$^{fl/fl}$* controls and was accompanied by a greater than two-fold increase in *Il6* expression (*Figure 5h*). These results are consistent with feeding-induced TCPTP fluctuations in POMC neurons altering responses to insulin to help coordinate hepatic glucose metabolism.

## TCPTP regulation of IR signaling alters POMC neuronal activation

Previous studies have shown that feeding/fasting can influence neuronal plasticity to coordinate energy homeostasis (*Nasrallah and Horvath, 2014*). As feeding-associated fluctuations in TCPTP serve to coordinate IR signaling in POMC and AgRP neurons (*Dodd et al., 2017*; *Dodd et al., 2015*; *Dodd et al., 2018*; *Tiganis, 2013*), we asked if the effects of TCPTP-deficiency on POMC neuronal



**Figure 5.** Feeding-induced repression of TCPTP in POMC neurons promotes the ICV insulin-induced repression of hepatic gluconeogenesis. (a, d) Experimental paradigm schematics. 8–9 week-old male C57BL/6 mice were *ad libitum* fed till satiated or food-restricted (just prior to lights out, 6:30pm) and administered ICV saline or insulin (0.1 mU/animal, 5 injections over 5 hr) as indicated and subjected 1 hr later to either (b, e) pyruvate tolerance tests (1 mg/g; areas under curves were determined and significance determined using a two-tailed t-test) or livers were extracted for (c, f) quantitative PCR and significance determined using a two-tailed t-test. 8–10 week-old POMC-TC or *Ptpn2*fl/fl male mice were food-restricted and administered insulin or vehicle ICV as indicated above and subjected to either (g) pyruvate tolerance tests (1 mg/g; areas under curves were determined and significance determined using a two-tailed t-test) or livers were extracted for (h) quantitative PCR and significance determined using a one-way ANOVA. Representative and quantified results are shown (means ± SEM) for the indicated number of mice.

DOI: https://doi.org/10.7554/eLife.38704.012

activity and glucose metabolism might be associated with alterations in IR signaling in POMC neurons. First, we assessed the influence of TCPTP-deficiency on insulin-induced PI3K/AKT signaling. To this end we crossed POMC-TC mice (on a *Pomc*-eGFP reporter background) onto the *Insr*fl/+ heterozygous background to generate POMC-TC-IR mice so that insulin signaling might approximate that in *Ptpn2*fl/fl controls. Neither TCPTP-deficiency, nor combined TCPTP deficiency and IR heterozygosity affected the number of ARC POMC neurons in adult mice (*Figure 6—figure supplement 1a*). Consistent with our previous studies (*Dodd et al., 2015*) we found that TCPTP deficiency enhanced insulin-induced ARC PI3K/AKT signaling in POMC neurons, as assessed by monitoring for AKT Ser-473 phosphorylation (p-AKT) in mice on the *Pomc*-eGFP reporter background. Importantly, we found that the enhanced PI3K/AKT signaling otherwise associated with TCPTP-deficiency was significantly repressed, albeit not completely corrected by IR heterozygosity (*Figure 6a–b*). To explore if the enhanced IR-dependent signaling in POMC-TC mice might impact on electrophysiological activity and the proportion of POMC neurons activated versus inhibited by insulin we recorded from

**Figure 6.** TCPTP regulation of insulin signaling alters POMC neuronal activation and the control of hepatic glucose metabolism. (a–b) 8–10 week-old male *Ptpn2^{fl/fl}*, POMC-TC or *Pomc*-Cre;*Ptpn2^{fl/fl}*;*Insr^{fl/+}* (POMC-TC-IR) male mice on the *Pomc*-eGFP background were fasted overnight and intraperitoneally administered insulin (2.5 mU/g, 15 min) and brains extracted for ARC p-AKT immunohistochemistry and pAKT positive POMC neurons quantified across the rostral-caudal extent of the hypothalamus and significance determined using a one-way ANOVA followed by Tukey multiple

*Figure 6 continued on next page*

*Figure 6 continued*

comparison test. (**c–d**) Whole-cell patch clamp recordings of POMC neurons from *Ptpn2*[fl/fl], POMC-TC or POMC-TC-IR mice on the *Pomc*-eGFP background in response to insulin (100 nM). (**c**) Grouped POMC population insulin responses, (**d**) membrane potential change and firing frequencies recorded from the entire rostral-caudal extent of the hypothalamic POMC neuronal population. Results shown are means ± SEM for the indicated number of cells in pie charts. Electrophysiological responses in n = 48 independent neurons were assessed across 8–10 mice and significance determined using a two-tailed t-test. 10–12 week-old male *Ptpn2*[fl/fl], POMC-TC or POMC-TC-IR mice were fasted overnight and brains either (**e**) microdissected and MBH processed for quantitative PCR, or (**f**) paraformaldehyde-fixed and processed for immunofluorescence microscopy monitoring for α-MSH in the PVH; the fluorometric integrated density of α-MSH staining in the PVH was quantified and significance determined using a one-way ANOVA followed by Tukey multiple comparison test. 8–10 week-old *Ptpn2*[fl/fl], POMC-TC or POMC-TC-IR overnight fasted male mice were administered ICV saline or insulin (0.1 mU/animal, 5 injections over 5 hr) and subjected 1 hr later to either (**g-i**) pyruvate tolerance tests (1 mg/g; areas under curves determined and statistical significance assessed using a two-way ANOVA followed by Tukey multiple comparison test) or livers were extracted for (**j**) quantitative PCR (significance assessed using a one-way ANOVA followed by Tukey multiple comparison test). In (**c**) values rounded up to the nearest integer. Representative and quantified results are shown (means ± SEM) for the indicated number of mice.

DOI: https://doi.org/10.7554/eLife.38704.013

The following figure supplements are available for figure 6:

**Figure supplement 1.** ARC POMC neurons in Ptpn2[fl/fl], POMC-TC and POMC-TC-IR mice.

DOI: https://doi.org/10.7554/eLife.38704.014

**Figure supplement 2.** TCPTP regulation of insulin signaling alters the POMC neuronal control of glucose metabolism.

DOI: https://doi.org/10.7554/eLife.38704.015

**Figure supplement 3.** Intracerebroventricularly (ICV) administered insulin does not alter body weight, adiposity and food intake.

DOI: https://doi.org/10.7554/eLife.38704.016

GFP-positive POMC neurons in coronal slices from POMC-TC-IR mice (on the *Pomc*-eGFP reporter background). IR heterozygosity significantly decreased the number of POMC neurons that were activated by insulin in POMC-TC mice, from 36% (12/33; *Figure 1g*; *Supplementary file 1*) to 17% (8/48; *Figure 6c*), so that the proportion of activated neurons in POMC-TC-IR mice was not significantly different from controls (*Supplementary file 1*). Excited neurons in POMC-TC-IR mice were significantly depolarised in response to insulin (baseline membrane potential −50.9 ± 3.74 mV; insulin −41.71 ± 3.66 mV) and firing rate increased (baseline 0.149 ± 0.26 Hz; insulin 1.141 ± 1.08 Hz), whereas insulin-inhibited neurons were significantly hyperpolarised (baseline membrane potential −41.56 ± 5.96 mV; insulin −49.31 ± 9.02 mV) and the firing rate reduced (baseline 0.891 ± 0.22 Hz; insulin 0.22 ± 0.10 Hz) in response to insulin (*Figure 6d*). Therefore, these results indicate that PTPN2 influences POMC neuronal electrical activity through the control of IR signalling.

Next, we determined whether the IR-mediated control of POMC neuronal electrical activity might influence the melanocortin response and hepatic glucose metabolism. POMC neuronal activation is accompanied by increased *Pomc* expression and the processing of POMC to α-MSH (*Schneeberger et al., 2013*); α-MSH is released from POMC neuronal terminals at the PVH where it drives melanocortin-dependent responses (*Varela and Horvath, 2012*). Therefore, we measured hypothalamic *Pomc* expression and stained for α-MSH in the PVH in *Ptpn2*[fl/fl], POMC-TC and POMC-TC-IR. We found that hypothalamic *Pomc* gene expression (*Figure 6e*) and α-MSH staining in the PVH were increased in POMC-TC mice and these were corrected by IR heterozygosity (*Figure 6f*). Moreover, we found that the enhanced insulin-POMC-mediated repression of hepatic gluconeogenesis (as assessed by the ability of ICV insulin to repress pyruvate-induced glucose excursions and hepatic gluconeogenic gene expression; *Figure 6g–i*) and improved glucose metabolism (as assessed in glucose, insulin and pyruvate tolerance tests and by measuring blood glucose levels; *Figure 6—figure supplement 2a–d*) in POMC-TC mice were largely corrected in POMC-TC-IR mice so that POMC-TC-IR mice more closely resembled *Ptpn2*[fl/fl] controls; ICV insulin did not affect body weight, food intake or adiposity in *Ptpn2*[fl/fl], POMC-TC or POMC-TC-IR mice (*Figure 6—figure supplement 3a–d*). Taken together these results are consistent with deficiencies in TCPTP promoting IR signaling to increase the proportion of POMC neurons that are activated by insulin to drive the melanocortin-dependent improvement in glucose metabolism.

## Elevated TCPTP in POMC neurons in obesity perturbs glucose metabolism

Our studies indicate that the inhibition of POMC neurons promotes hepatic gluconeogenesis and whole-body insulin resistance, whereas the activation of POMC neurons represses HGP. Accordingly, we reasoned that the increase in TCPTP in POMC neurons in obesity and the consequent shift in POMC neuronal responses (so that a greater proportion of POMC neurons are inhibited by insulin) might promote HGP and perturb glucose metabolism and contribute to the development of type 2 diabetes in obesity. To explore this, we determined if the deletion of TCPTP in POMC neurons in obese mice might enhance the insulin-POMC-mediated repression of hepatic gluconeogenesis and improve glucose metabolism. To test this, we ICV administered insulin to *Ptpn2*[fl/fl] or POMC-TC mice that had been fed a high fat diet for 12 weeks and subjected mice to pyruvate tolerance tests (*Figure 7a–d*); TCPTP-deficiency in POMC neurons did not alter body weight or adiposity (*Figure 7—figure supplement 1a–b*). Whereas control mice remained unresponsive to ICV insulin, TCPTP deficiency in POMC neurons repressed pyruvate-induced glucose excursions in response to ICV insulin and this was accompanied by increased hepatic *Il6* expression and STAT-3 signaling and the repression of *Pck1* and *G6pc* expression (*Figure 7a–d*). These results are consistent with TCPTP deficiency enhancing the ICV insulin-induced repression of HGP. To examine whether this might influence whole-body glucose metabolism, we first measured blood glucose and plasma insulin levels and subjected mice to insulin and glucose tolerance tests. As noted previously (*Dodd et al., 2015*), TCPTP deficiency in POMC neurons did not alter glucose excursions in insulin and glucose tolerance tests that primarily measure the abilities of skeletal muscle and adipose tissue to clear glucose (*Figure 7—figure supplement 1c–d*). However, blood glucose and plasma insulin levels in 12 hr fasted mice were reduced, consistent with improved whole-body insulin sensitivity (*Figure 7e–f*). To further explore this, we subjected 12 week high fat fed mice to hyperinsulinemic euglycemic clamps. Glucose infusion rates were significantly increased in POMC-TC mice (*Figure 7g*; *Figure 7—figure supplement 1e*), consistent with enhanced systemic insulin sensitivity. Although glucose disappearance (Rd) was unaltered (*Figure 7h*), endogenous glucose production was suppressed in POMC-TC mice (*Figure 7i*) and accompanied by reduced hepatic gluconeogenic (*Pck1* and *G6pc*) gene expression (*Figure 7j*). Therefore, TCPTP-deficiency in POMC neurons in obesity improves glucose metabolism through the repression of HGP. Taken together our findings are consistent with elevated TCPTP in obesity perturbing POMC insulin responses and the proportion of POMC neurons activated versus inhibited by insulin to drive HGP and fasting hyperglycemia in obesity.

## Discussion

In this study we demonstrate that POMC neurons can be either inhibited or activated by insulin (or otherwise not respond) and that the balance of excitation versus inhibition is dictated by the phosphatase TCPTP. Moreover, we demonstrate that TCPTP may orchestrate these effects through the regulation of IR signaling to coordinate the melanocortin response and hepatic glucose metabolism in response to feeding and fasting. This provides an elegant system for integrating the melanocortin circuit with the nutritional state of the organism and coordinating feeding with the CNS control of glucose metabolism.

The IR is widely expressed in the hypothalamus and is present in both POMC and AgRP neurons (*Havrankova et al., 1978*; *Könner et al., 2007*). Although its role in AgRP neurons and HGP has been firmly established (*Könner et al., 2007*), its importance in POMC neurons and glucose metabolism has received less attention, as early studies showed that IR deletion in POMC neurons had no effect on HGP and insulin sensitivity (*Könner et al., 2007*). Our studies suggest that at least one explanation for this could be that any effect on glucose metabolism would be negated by the concomitant deletion of IR in POMC neurons that are excited and inhibited by insulin. Moreover, our studies indicate that any response of POMC neurons to insulin would be dictated by the nutritional state of the organism. C57BL/6 mice feed predominantly at night during the first four hours of the dark cycle (*Dodd et al., 2017*). Previous studies addressing the role of IR deletion in POMC neurons would have been conducted during the day (*Könner et al., 2007*; *Shin et al., 2017*), when mice are effectively fasted and TCPTP levels are elevated (*Dodd et al., 2017*) and/or in overnight fasted mice (*Könner et al., 2007*). Under these conditions, the TCPTP-mediated suppression of insulin signaling would render POMC neurons largely unresponsive or inhibited by insulin and antagonise the CNS



**Figure 7.** Elevated TCPTP in POMC neurons in obesity promotes HGP and systemic insulin resistance. (a) Experimental paradigm schematic. (b) *Ptpn2*<sup>fl/fl</sup> or POMC-TC male mice were high fat fed (HFF) for 12 weeks and after an overnight fast ICV administered saline or 0.1 mU/animal (5 x injections over 5 hr) and subjected 1 hr later to either (b) a pyruvate tolerance test (1.5 mg/g) or livers extracted for (c) quantitative PCR or (d) immunoblotting; areas under curves were determined and significance determined using a one-way ANOVA followed by Tukey multiple comparison test, or a two-way ANOVA followed by Sidak multiple comparison test. *Ptpn2*<sup>fl/fl</sup> or POMC-TC male mice were HFF for 12 weeks and fed and fasted (e) blood glucose and (f) plasma insulin levels were assessed; significance was determined using a two-way ANOVA followed by Sidak multiple comparison test. *Ptpn2*<sup>fl/fl</sup> or POMC-TC male mice were HFF for 12–14 weeks and subjected to hyperinsulinemic-euglycemic conscious clamps. (g) Glucose infusion rates (GIR; analysed by two-way ANOVA with repeated measures followed by Sidak multiple comparison test), (h) glucose disappearance rates (Rd) and (i) basal and clamped endogenous glucose production (EGP; glucose appearance rate minus GIR) were determined and (j) livers extracted for quantitative PCR (analysed using a two-tailed t-test or a two-way ANOVA followed by Sidak multiple comparison test). Representative and quantified results are shown (means ± SEM) for the indicated number of mice.

DOI: https://doi.org/10.7554/eLife.38704.017

The following figure supplement is available for figure 7:

**Figure supplement 1.** TCPTP deletion in POMC neurons in obesity represses HGP and improves whole-body glucose homeostasis.
DOI: https://doi.org/10.7554/eLife.38704.018

insulin-mediated repression of HGP. Therefore, it is likely that the responses seen in our studies would have been missed by previous studies. Consistent with this assertion we found that ICV insulin repressed HGP in fed mice, but not in fasted mice. In addition, Steculorum et al. (*Steculorum et al., 2016*) have reported that the acute activation of AgRP neurons using DREADDs has no effect on HGP, but represses brown adipose tissue glucose uptake and insulin sensitivity, whereas the acute activation of POMC neurons has no discernible influence on glucose metabolism. Although this is consistent with the acute CNS regulation of glucose metabolism being independent of the melano-cortin circuit, our studies demonstrate that the chronic inhibition or activation of POMC neurons, via DREADDs, or deletion of the phosphatase TCPTP, profoundly influences hepatic insulin sensitivity and glucose metabolism. The chronic activation or inhibition of POMC neurons decreased or increased hepatic gluconeogenesis and was accompanied by the promotion or repression of hepatic *Il6* expression and STAT3 signaling and the repression or induction of gluconeogenic gene expression respectively. The impact of the pharmacogenetic inhibition of POMC neurons with DREADDs on hepatic glucose metabolism was substantiated in hyperinsulinemic euglycemic clamps where endogenous glucose production was markedly increased and systemic insulin sensitivity decreased. Conversely TCPTP deletion in POMC neurons resulted enhanced IR signaling and POMC neuronal excitation to repress HGP and significantly enhanced hepatic and whole-body insulin sensitivity. Moreover, the acute activation of AgRP neurons may elicit melanocortin-independent effects on brown adipose tissue glucose uptake (*Steculorum et al., 2016*), our recent studies indicate that TCPTP-deletion in AgRP enhances the insulin-mediated inhibition of AgRP neurons (*Dodd et al., 2017*) and the IR-dependent repression of HGP (*Dodd et al., 2018*). Therefore, the chronic activation of POMC neurons and concomitant inhibition of AgRP neurons by insulin would drive melano-cortin-dependent responses to modulate HGP and glucose homeostasis. This may allow for a graded response to changing nutritional states, with acute changes in insulin acting via AgRP-dependent but melanocortin-independent pathways to affect BAT glucose uptake, and the sustained hyperglycemia and hyperinsulinemia accompanying more intense caloric intake additionally engaging the melanocortin circuit and the liver to allow for a concerted systemic response to lower post-prandial blood glucose levels.

Our studies indicate that the abundance of the IR phosphatase TCPTP determines whether POMC neurons are excited or inhibited by insulin and controls the intensity of IR signaling. TCPTP deletion in POMC neurons (marked by the *Pomc*-eGFP reporter), or inhibition with the TCPTP selective inhibitor compound 8, shifted POMC neural responses so that the majority of POMC neurons were activated by insulin. Our recent studies have shown that TCPTP abundance in AgRP and POMC neurons is subject to diurnal fluctuations linked to feeding (*Dodd et al., 2017*). Increases in circulating corticosterone that accompany fasting drive hypothalamic TCPTP expression (*Dodd et al., 2017*). By contrast, feeding inhibits TCPTP expression and promotes its rapid degradation via the proteasome (*Dodd et al., 2017*). We have reported that in AgRP neurons this serves to coordinate the browning and thermogenic activity of white fat with the expenditure of energy linked to feeding (*Dodd et al., 2017*). In other studies we have shown that such fluxes in TCPTP in AgRP neurons also influence hepatic glucose metabolism (*Dodd et al., 2018*), which is in keeping with the previously established role for the IR in AgRP neurons in glucose metabolism (*Könner et al., 2007*). Although we did not assess the influence of feeding and fasting on the activation of POMC neurons by insulin, we demonstrated that insulin administered ICV repressed hepatic gluconeogenesis in fed mice, when TCPTP is repressed (*Dodd et al., 2017*), but not in food restricted or fasted mice, when TCPTP expression is increased (*Dodd et al., 2017*). Importantly we found that deleting TCPTP in POMC neurons, which enhanced IR signaling and resulted in a recruitment of POMC neurons activated by insulin, reinstated the ICV insulin-induced repression of gluconeogenesis and hepatic glucose output in otherwise unresponsive fasted mice. In keeping with this assertion, we found that chow-fed mice lacking TCPTP in POMC neurons exhibited a marked improvement in insulin sensitivity (as assessed in hyperinsulinemic euglycemic clamps) that was accounted for by a reduction in endogenous glucose production (a measure of HGP) and decreased gluconeogenesis, as assessed in pyruvate tolerance tests and the decreased expression of hepatic gluconeogenic genes. The improvement in glucose homeostasis was corrected in POMC-TC-IR mice, where the enhanced insulin-induced PI3K/AKT signaling in mice lacking TCPTP in POMC neurons was corrected to that of control mice. This would suggest that TCPTP deletion, or decreased TCPTP in POMC neurons in response to feeding, represses HGP by enhancing IR signaling and permitting POMC neurons to be

activated by insulin. A caveat of these findings is that early in development *Pomc*-Cre is active in roughly 25% of NPY neurons (*Padilla et al., 2010*; *Xu et al., 2018*), so that the improved glucose metabolism in POMC-TC mice might at least in part reflect the enhancement of insulin signalling in AgRP/NPY neurons. However, given that (i) TCPTP abundance in POMC neurons (marked by the *Pomc*-eGFP reporter) correlated with insulin-induced electrophysiological responses, (ii) TCPTP inhibition in POMC neurons (marked by the *Pomc*-eGFP) increased the proportion of activated neurons, (iii) α-MSH staining in the PVH was increased in POMC-TC mice, and iv) DREADD-mediated activation of POMC neurons repressed HGP as seen in POMC-TC mice, our findings are consistent with feeding-associated TCPTP fluxes in POMC neurons coordinating POMC neuronal activity and thereby hepatic glucose metabolism in response to feeding.

Precisely how TCPTP influences POMC neuronal electrophysiological responses to insulin remains to be determined. One mechanism could involve TCPTP affecting synaptic plasticity by influencing pre-synaptic excitatory and inhibitory inputs, including GABAergic inputs (*Cowley et al., 2001*; *Dietrich and Horvath, 2013*; *Horvath et al., 2010*; *Pinto et al., 2004*; *Sternson et al., 2005*; *Vong et al., 2011*; *Yang et al., 2011*). However, our studies indicate that there is no overt change in relative excitatory and inhibitory inputs into POMC perikarya (Horvath *et al.*, unpublished observation). Another mechanism could involve IR-dependent pathways that directly influence the activation of TRPC channels that depolarise POMC neurons (*Qiu et al., 2010*), or inwardly rectifying K$^+$ channels that inhibit POMC neurons (*Plum et al., 2006*). Yet another mechanism could involve TCPTP altering mitochondrial dynamics, as mitochondrial fusion in AgRP neurons drives ATP production and excitation (*Dietrich et al., 2013*; *Nasrallah and Horvath, 2014*), whereas mitochondrial fusion in POMC neurons drives α-MSH production and the melanocortin response (*Schneeberger et al., 2013*). In cardiomyocytes, insulin-induced PI3K signaling promotes mitochondrial fusion (*Parra et al., 2014*), but it remains to be established if insulin similarly alters mitochondrial dynamics in POMC neurons and whether TCPTP influences POMC neuron electrophysiologial responses through the IR-mediated control of mitochondrial fusion. Irrespective, our studies define an exquisite mechanism by which to link feeding and fasting and the associated changes in insulin levels and glucocorticoids that drive TCPTP expression with the control of POMC neuronal excitability and the coordination of hepatic glucose metabolism for the maintenance of euglycemia. Our studies point towards this glucoregulatory mechanism being defective in obesity. The increased hypothalamic TCPTP in obesity probably occurs as a consequence of the heightened glucocorticoid and leptin levels that we have shown previously to drive TCPTP expression in the ARC (*Dodd et al., 2017*; *Loh et al., 2011*). Irrespective, our studies point towards the heightened hypothalamic TCPTP levels in obesity resulting in the majority of POMC neurons being inhibited by insulin, to contribute to the increased HGP and fasting hyperglycemia that accompany the obese state. In humans, peripheral insulin resistance in obesity has been linked to CNS insulin resistance (*Anthony et al., 2006*; *Hirvonen et al., 2011*; *Tschritter et al., 2006*). Moreover the ability of intranasally administered insulin to elicit effects in the hypothalamus and suppress HGP and promote glucose uptake in lean individuals is defective in overweight individuals (*Heni et al., 2017*; *Heni et al., 2014*). Therefore, we propose that perturbations in glucose metabolism in obesity, may at least in part be due to TCPTP repressing the IR-dependent activation of POMC neurons and the overall melanocortin output response.

## Materials and methods

### Key resources table

| Reagent type (species) or resource | Designation | Source or reference | Identifiers | Additional information |
|---|---|---|---|---|
| Strain, strain background (Mouse, *Pomc*-Cre, C57BL/6J background) | B6.FVB-Tg(Pomc-cre) 1Lowl/J, C57BL/6J | PMID: 17556551 | RRID:IMSR_JAX:010714 | |

*Continued on next page*

*Continued*

| Reagent type (species) or resource | Designation | Source or reference | Identifiers | Additional information |
|---|---|---|---|---|
| Strain, strain background (Mouse, Insr$^{fl/fl}$, C57BL/6J background) | B6.129S4)-Insr$^{tm1Khn}$/J, C57BL/6J | PMID: 9844629 | RRID:IMSR_JAX:006955 | |
| Strain, strain background (Mouse, *Pomc*-eGFP, C57BL/6J background) | C57BL/6J-Tg (Pomc-EGFP)1Low/J, C57BL/6J () | PMID: 11373681 | RRID:IMSR_JAX:009593 | |
| Strain, strain background (Mouse, Z/EG, C57BL/6J background) | Tg(CAG-Bgeo/GFP) 21Lbe/J | PMID: 11105057 | RRID:IMSR_JAX:003920 | |
| Strain, strain background (Mouse, Ptpn2$^{lox/lox}$, C57BL/6J background) | Mouse (*Ptpn2$^{fl/fl}$*) | PMID: 22590589 | N/A | |
| Antibody (4970) | Rabbit monoclonal anti β-Actin | Cell Signaling Technology | RRID:AB_2223172 | (1:2000) |
| Antibody (AM1403031) | Mouse polyclonal anti Gapdh | Ambiom | AM1403031 | (1:50000) |
| Antibody (9145) | Rabbit monoclonal anti-p-STAT3 (Y705) | Cell Signaling Technology | RRID: AB_10694804 | (1:1000) |
| Antibody (9139) | Mouse monoclonal anti STAT3 | Cell Signaling Technology | RRID:AB_331757 | (1:1000) |
| Antibody (4060) | Rabbit monoclonal anti-p-AKT (Ser-473) | Cell Signaling Technology | RRID:AB_2315049 | (1:2000 (WB), 1:300 (IHC)) |
| Antibody (MM0019p) | Mouse monoclonal anti TCPTP | Medimabs | MM0019p | (1:1000 (WB), 1:200 (IHC)) |
| Antibody (ab13970) | Chicken polyclonal anti GFP | abcam | RRID:AB_300798 | (1:1000) |
| Antibody (PHH02930) | Rabbit polyclonal anti POMC | Phoenix Pharmaceuticals | RRID:AB_2307442 | (1:1000) |
| Antibody (SC-711) | Rabbit polyclonal anti IR | Santa Cruz | RRID:AB_631835 | (1:2000) |
| Antibody (44–804G) | Rabbit polyclonal anti p-IR (Tyr-1162, 1163) | ThermoScientific | RRID:AB_2533762 | (1:2000) |
| Antibody (sc-52) | Rabbit polyclonal anti c-Fos | Santa Cruz | RRID:AB_2106783 | (1:4000) |
| Antibody (632496) | Rabbit polyclonal anti dsRed | Clontech | RRID:AB_10013483 | (1:2500) |
| Antibody (AS597) | guinea pig anti-alpha-MSH | Antibody Australia | AS597 | (1:1000) |
| Antibody (9272) | Rabbit polyclonal anti Akt | Cell Signaling Technology | RRID:AB_329827 | (1:5000) |
| Commercial assay or kit (RI-13k) | Rat Insulin radioimmunoassay | Millipore | RI-13k | |
| Chemical compound, drug (3435) | Insulin (human) recombinant | Tocris Bioscience | 3435 | |
| Chemical compound, drug (C0832) | Clozapine N-oxide | Sigma-Aldrich | C0832 | |

*Continued on next page*

*Continued*

| Reagent type (species) or resource | Designation | Source or reference | Identifiers | Additional information |
|---|---|---|---|---|
| Chemical compound, drug (169625) | Actrapid (human, Insulin) | Novo Nordisk Pharmaceuticals | 169625 | |
| Sequence-based reagent (Mm00435874_m1) | *Pomc* TaqMan Gene Expression Assay | ThermoFisher | Mm00435874_m1 | |
| Sequence-based reagent (Mm03048253_m1) | *Npy* TaqMan Gene Expression Assay | ThermoFisher | Mm03048253_m1 | |
| Sequence-based reagent (Mm00475829_g1) | *Agrp* TaqMan Gene Expression Assay | ThermoFisher | Mm00475829_g1 | |
| Sequence-based reagent (Mm99999915_g1) | *Gapdh* TaqMan Gene Expression Assay | ThermoFisher | Mm99999915_g1 | |
| Sequence-based reagent (Mm01247058_m1) | *Pck1* TaqMan Gene Expression Assay | ThermoFisher | Mm01247058_m1 | |
| Sequence-based reagent (Mm00839363_m1) | *G6p* TaqMan Gene Expression Assay | ThermoFisher | Mm00839363_m1 | |
| Sequence-based reagent (Mm00446190_m1) | *Il6* TaqMan Gene Expression Assay | ThermoFisher | Mm00446190_m1 | |

## Animals

We maintained mice on a 12 hr light-dark cycle in a temperature-controlled high-barrier facility (Monash ARL) with free access to food and water as per NHMRC Australian Code of Practice for the Care and Use of Animals. The *Pomc*-Cre [from from B. Lowell, Beth Israel Deaconess Medical Center, Boston, Massachusetts (*Balthasar et al., 2004*)], *Pomc*-eGFP [C57BL/6J-Tg(Pomc-EGFP)1Low/J, Jackson Laboratory (*Cowley et al., 2001*), Z/EG [Tg(CAG-Bgeo/GFP)21Lbe/J, Jackson Laboratory] and *Ptpn2*$^{fl/fl}$ (*Loh et al., 2011*) were on a C57BL/6J background. To generate *Pomc*-Cre;*Ptpn2*$^{fl/fl}$ (POMC-TC), *Ptpn2*$^{fl/fl}$ (C57BL/6) mice were bred with *Pomc*-Cre (C57BL/6) mice. To generate *Pomc*-Cre;*Ptpn2*$^{fl/fl}$;Z/EG mice Pomc-TC mice were mated with Z/EG reporter mice. POMC-TC mice were mated with *Insr1*$^{fl/fl}$ mice (*Brüning et al., 1998*) to generate *Pomc*-Cre;*Ptpn2*$^{fl/fl}$;*Insr*$^{fl/+}$ (POMC-TC-IR) mice. To generate *Ptpn2*$^{fl/fl}$;*Pomc*-eGFP, *Pomc*-Cre;*Ptpn2*$^{fl/fl}$;*Pomc*-eGFP or *Pomc*-Cre;*Ptpn2*$^{fl/fl}$; *Insr*$^{fl/+}$;*Pomc*-eGFP mice, *Ptpn2*$^{fl/fl}$, POMC-TC or POMC-TC-IR mice were mated with *Pomc*-eGFP mice respectively. Mice were fed a standard chow (8.5% fat; Barastoc, Ridley AgriProducts, Australia) or a high-fat diet (23% fat; 45% of total energy from fat; SF04-027; Specialty Feeds) as indicated. Experiments were approved by the Monash University School of Biomedical Sciences Animal Ethics Committee (MARP2013/137).

## Immunohistochemistry

For brain immunohistochemistry, mice were anaesthetized and perfused transcardially with heparinized saline [10,000 units/l heparin in 0.9% (w/v) NaCl] followed by 4% (w/v) paraformaldehyde in phosphate buffer (0.1 M, pH 7.4). Brains were post-fixed overnight and then kept for four days in 30% (w/v) sucrose in 0.1 M phosphate buffer to cryoprotect the tissue, before freezing on dry ice. 30 μm sections (120 μm apart) were cut in the coronal plane throughout the entire rostral-caudal extent of the hypothalamus.

For detection of eGFP and TCPTP or POMC and mCherry sections were subjected to antigen retrieval in citrate acid buffer [10 mM Sodium Citrate, 0.05% (v/v) Tween 20, pH 6.0] at 85°C for 40 min.

Sections were incubated at room temperature for 1 hr in blocking buffer [0.1 M phosphate buffer, 0.2% (v/v) Triton X-100, 10% (v/v) normal goat serum (Sigma, St. Louis, MO); TCPTP staining

blocking buffer contained unlabelled mouse IgG (1:500, Vector, Burlingame, CA)] and then overnight at 4°C in chicken anti-eGFP (1/1000; ab13970, Abcam, Cambridge, UK), rabbit anti-dsRed (1:2500, Clontech, CA), mouse anti-TCPTP (1/200; 6F3 from Medimabs, Quebec, Canada) or rabbit anti-POMC (1/1000; H-029–30, Phoenix Pharmaceuticals, Burlingame, CA), in blocking buffer. After washing with PBS, sections were incubated with goat anti-chicken Alexa-Fluor 488, goat anti-rabbit Alexa-Fluor-488-, goat anti-rabbit Alexa-Fluor-568- or goat anti-mouse Alexa-Fluor-568-conjugated secondary antibodies (1/1000, Life Technologies, VIC, Australia) in blocking buffer for 2 hr at room temperature. Sections were mounted with Mowiol 4–88 mounting media and visualized using an Olympus Provis AX70 microscope. Images were captured with an Olympus DP70 digital camera and processed using ImageJ software (NIH, MA).

For immunohistochemical detection of α-MSH, 10–12 week-old *Ptpn2<sup>fl/fl</sup>*, POMC-TC or POMC-TC-IR male were overnight fasted. Mice were transcardially perfused and the brains were post-fixed and cut in the coronal plane throughout the entire rostral-caudal extent of the hypothalamus (as described above). Sections were incubated at room temperature for 1 hr in blocking buffer and then overnight at 4°C in guinea pig anti-α-MSH (1/1000; AS597, Antibody Australia) in blocking buffer. After washing with PBS, sections were incubated with goat anti- guinea pig Alexa-Fluor-488-conjugated secondary antibody (1/1000, Life Technologies, VIC, Australia) in blocking buffer for 2 hr at room temperature. Sections were mounted, visualized and captured as described above. To determine the fluorometric integrated density of α-MSH staining in the PVH, a region of interest was first drawn around the PVH using ImageJ software. All images underwent skeletonization using a consistent threshold setting and integrated density was quantified using ImageJ software.

For immunohistochemical detection of biocytin, GFP and TCPTP in ex vivo whole slices used for patch clamp electrophysiology, recorded POMC neurons were filled with biocytin (10 mM, Sigma) using the patch clamp pipette. Slices were fixed for 1 hr in 4% (w/v) paraformaldehyde in phosphate buffer (0.1 M, pH 7.4) and rinsed three times in Tris-Triton buffer (TBS-T, 50 mM Tris-HCl, 0.9% (w/v) NaCl, 1% (w/v) Triton X-100, pH 7.5). Sections were subjected to antigen retrieval in citrate acid buffer at 85°C for 40 min. Sections were incubated at room temperature for 1 hr in blocking buffer [0.1 M phosphate buffer, 1% (v/v) Triton X-100, 5% (v/v) normal goat serum and incubated overnight at 4°C in chicken anti-eGFP (1/1000) and mouse anti-TCPTP (1/100) in blocking buffer. After washing with TBS-T, sections were incubated with goat anti-chicken Alexa-Fluor-488-, goat anti-mouse Alexa-Fluor-568- and streptavidin-Alexa-700-conjugated secondary antibodies (1/1000, Life Technologies, VIC, Australia) in blocking buffer for 2 hr at room temperature. Sections were mounted with Mowiol 4–88 mounting media and visualized using an Olympus Provis AX70 microscope. Images were captured with an Olympus DP70 digital camera and processed using ImageJ software (NIH, MA).

## c-Fos and p-AKT immunohistochemistry

8–10 week-old male *Ptpn2<sup>fl/fl</sup>*, POMC-TC or POMC-TC-IR mice fed chow or high fat diets were overnight fasted and injected intraperitoneally with either vehicle or human insulin (0.85, 2.5 or 5 mU mU/g, SIGMA, St Louis, MO) for c-Fos or p-AKT (Ser-473) immunohistochemistry respectively. Mice were transcardially perfused (as described above) either 15 min (p-Akt staining) or 90 min (c-Fos staining) post-injection with a solution of 4% (w/v) paraformaldehyde. The brains were post-fixed overnight and then kept for two days in 30% (w/v) sucrose in 0.1 M phosphate buffer to cryoprotect the tissue, before freezing on dry ice. 30 μm sections (120 μm apart) were cut in the coronal plane throughout the entire rostral-caudal extent of the hypothalamus. For p-AKT immunostaining, sections were pre-treated for 20 min in 0.5% (w/v) NaOH and 0.5% (v/v) $H_2O_2$ in PBS, followed by immersion in 0.3% (w/v) glycine for 10 min. Sections were then placed in 0.03% (w/v) SDS for 10 min and placed in 4% (v/v) normal goat serum plus 0.4% (w/v) Triton X-100 for 20 min before incubation for 48 hr with rabbit anti-p-Akt (1:300; #4060, Cell Signaling Technology). For c-Fos immunostaining, sections were incubated at room temperature for 1 hr in blocking buffer [0.1 M phosphate buffer, 0.2% (v/v) Triton X-100, 10% (v/v) normal goat serum) and then overnight at 4°C in rabbit c-Fos antibody (1:4000, sc-52, Santa Cruz, CA, USA) in 1% (v/v) blocking buffer. For both p-AKT and c-Fos immunohistochemistry, sections were incubated for 2 hr at room temperature with goat anti-rabbit Alexa-Fluor-568-conjugated secondary antibody in 5% (v/v) blocking buffer. Sections were mounted with Mowiol 4–88 mounting media and visualized using an Olympus Provis AX70 microscope.

## Electrophysiological recordings

*Ad libitum* fed adult (>10 weeks of age) mice (between 9 am and 11 am) were anaesthetised with 2% (v/v) isoflurane, brains removed and coronal slices (250 µm) cut, using a vibrating blade microtome (Leica VTS1000), in cold (<4°C) carbogenated (95% $O_2$/5% $CO_2$) aCSF of the following composition: 127 mM NaCl, 1.2 mM $KH_2PO_4$, 1.9 mM KCl, 26 mM $NaHCO_3$, 3 mM D-glucose, 7 mM manitol, 2.4 mM $CaCl_2$, 1.3 mM $MgCl_2$). Once cut, slices were then heated for 20 min at 34°C and incubated at room temperature prior to recording.

Slices were transferred to a recording chamber and continuously perfused with aCSF. *Pomc*-eGFP neurons in the arcuate nucleus were identified and visualised using an Axioskop FS2 (Zeiss) microscope fitted with DIC optics, infrared videomicroscopy, florescence and a GFP filter set. Patch pipettes were pulled from thin-walled borosilicate glass (GC150-TF10, Harvard Apparatus) with resistances between 3 and 8 MΩ when filled with intracellular solution of the following composition: 140 mM K-gluconate, 10 mM KCl, 10 mM HEPES, 1 mM EGTA and 2 mM $Na_2ATP$) with osmolality and pH adjusted with sucrose and KOH, respectively. Whole-cell recordings were made with Axopatch 1D and Multiclamp 700B amplifiers (Molecular Devices LLC, Sunnyvale, CA, USA). Current and voltage data were filtered at 2–5 kHz and 1 kHz, respectively. For data analysis signals were digitized at 2–10 kHz, and stored on a personal computer running pClamp9 or 10 software (Axon Instruments). Resting membrane potential is expressed as the read-out from the amplifier and was not corrected for the liquid junction potential offset (11 mV). Recombinant human insulin (#3435, Tocris Bioscience, UK) was made up as a stock solution and diluted to 100 nM in aCSF immediately before electrophysiological recording.

In TCPTP inhibitor experiments, 8–12 week old chow fed or 8 week high fat fed *Pomc*-eGFP mice (20 week old) were cannulated into the lateral ventricle (as described below) and ICV administered vehicle or TCPTP inhibitor [Compound 8 (*Zhang et al., 2009*), 2 nmol/animal/day in 1.5 µl], every day for three days prior to the start of patch clamp electrophysiological experiments. TCPTP inhibitor (20 nM) was also included in the intracellular pipette solution. Resting membrane potential is expressed as the read-out from the amplifier and was not corrected for the liquid junction potential offset (11 mV).

## Hyperinsulinaemic euglycaemic clamps

For unconscious hyperinsulinemic euglycemic clamps, 8–10 week old POMC-TC and *Ptpn2^{fl/fl}* mice were fasted for 4 hr and anesthetized on the morning of the experiment with an intraperitoneal injection of sodium pentobarbitone (100 mg/kg). A catheter was inserted into the right jugular vein for tracer infusion and another catheter inserted into the left carotid artery for sampling. A tracheostomy was also performed to prevent upper respiratory tract obstruction and body temperature maintained using a heat lamp. A primed (2 min, 3 µCi/min) continuous infusion (0.15 µCi/min) of [6-$^3$H]-glucose (GE Healthcare, Buckinghamshire, UK) was administered to measure whole body glucose turnover, as described previously (*Lamont et al., 2006*). Blood glucose was maintained at basal levels by the infusion of a 5% (w/v) glucose solution. Blood samples were collected during steady state conditions (rate of glucose appearance = rate of glucose disappearance) at 90, 100 and 110 min. The rate of glucose disappearance (Rd) was calculated by dividing the infusion rate of [6-$^3$H]-glucose (dpm/min) by the plasma [6-$^3$H]-glucose specific activity. The rate of whole-body glucose production was measured as the difference between the calculated Rd and the rate of infused glucose. Immediately following the collection of the last blood sample, animals were culled and extracted tissues were frozen in liquid nitrogen and stored at –70°C for subsequent analysis.

For hyperinsulinemic euglycemic clamps in conscious mice, 8-week-old *Pomc*-Cre mice injected one week prior with rAAV-hSyn-DIO-hM4D(Gi)-mCherry bilaterally into the ARC of the hypothalamus, or 20 week old *Ptpn2^{fl/fl}* and POMC-TC male mice that had been high fat fed for 12 weeks were anesthetized under 2% (v/v) isoflurane in 250 ml/min oxygen and the left common carotid artery and the right jugular vein catheterized for sampling and infusions, respectively, as previously described (*Fueger et al., 2007*). Lines were kept patent by flushing daily with 10–40 µl saline containing 200 units/ml heparin and 5 mg/ml ampicillin. Animals were housed individually after surgery and body weights recorded daily. Mice were allowed to recover for 4–5 days. One the day of the experiment mice were *ad libiutm* fed until 11 pm, food was then removed, and mice received a first injection of vehicle or CNO (0.3 mg/kg) and another injection 3 hr later (2 am). At 3 am a primed (2

min, 0.5 µCi/min) continuous infusion (0.05 µCi/min) of [3-³H]-glucose was administered to measure whole-body glucose turnover, as described previously (*Fueger et al., 2007*). At 4 am mice were received a continuous insulin infusion (4 mU/kg/min), and blood glucose was maintained at basal levels by a variable infusion of a 50% (w/v) glucose solution. For 20 week old *Ptpn2^{fl/fl}* and POMC-TC male mice that had been high fat fed for 12 weeks, food was removed at 8 am and at 11:30 am all mice received a primed (2 min, 0.5 mCi/min) continuous infusion (0.05 mCi/min) of 3-³H]-glucose to measure whole-body glucose turnover. At 1 pm mice received a continuous insulin infusion (4 mU/kg/min), and blood glucose was maintained at basal levels by a variable infusion of a 50% (w/v) glucose solution. For all experiments arterial blood samples were collected during steady state conditions (rate of glucose appearance = rate of glucose disappearance), and at 80, 90, 100, 110, and 120 min for determination of Rd and Ra as described above. Immediately following the collection of the last blood sample, animals were culled and extracted tissues were frozen in liquid nitrogen and stored at –70°C for subsequent analysis.

## DREADDs
8-week-old *Pomc*-Cre mice (on the *Pomc*-eGFP reporter background) were sterotaxically injected with rAAV-hSyn-DIO-hM4D(Gi)-mCherry or rAAV-hSyn-DIO-hM3D(Gq)-mCherry (UNC Vector Core, NC, *Krashes et al., 2011*) bilaterally into the ARC (coordinates, bregma: anterior-posterior, –1.40 mm; dorsal-ventral, –5.80 mm; lateral,±0.20 mm, 200 nl/side) as described previously (*Dodd et al., 2015*). DREADD's were activated by injection of CNO (0.3 mg/kg intraperitoneal, Sigma).

## Lateral ventricle cannulations
Under 2% (v/v) isoflurane in 250 ml/min oxygen 8–10 week-old chow *Ptpn2^{fl/fl}*, POMC-TC or POMC-TC-IR male mice or 12 week high fat fed *Ptpn2^{fl/fl}* and POMC-TC male mice were implanted stereotaxically with guide cannulas into the right lateral ventricle (0.2 mm posterior, 1.0 mm lateral from bregma). The tip of the guide cannula was positioned 1 mm above the injection site (1 mm ventral to the surface of the skull). All mice were allowed 4–5 days recovery before experimental manipulation. Where indicated, mice were fasted overnight and were administered ICV vehicle (PBS) or insulin (0.1 mU or 0.5 mU/animal in a volume of 2 µl) at 9 am, 10 am, 11 am, 12 am and 1 am. 2 hr later (3 pm) mice were subjected to either pyruvate tolerance tests or hypothalami and livers extracted for quantitative real time PCR.

## Metabolic measurements
Unless otherwise stated, insulin tolerance tests, glucose tolerance tests and pyruvate tolerance tests were performed on 4 hr, 6 hr and 6 hr fasted conscious mice respectively by injecting human insulin (0.5–0.65 mU insulin/g body weight), D-glucose (2 mg/g body weight), or sodium pyruvate (0.75–1.5 mg/g body weight) into the peritoneal cavity and measuring glucose in tail blood immediately before and at 15, 30, 45, 60, 90 and 120 min after injection using a Accu-Check glucometer (Roche, Germany). Fed and fasted (12 hr fast) plasma insulin levels were determined using a Rat insulin RIA kit (Merck Millipore, CA) or an in-house ELISA (Monash Antibody Technologies Facility) and blood glucose levels were determined using Accu-Check glucometer. The areas under glucose excursion curves were determined and expressed as mmol/l x min.

## Food intake measurements
Diurnal feeding was assessed using BioDAQ E2 cages (Research Diets, NJ) in 8-week-old male C57BL/6 mice. Food intake was grouped into 15 min time bins and smoothed over a 24 hr time period.

## Immunoblotting
Mouse tissues were dissected and immediately frozen in liquid $N_2$. For mediobasal hypothalamic micro-dissections, brains were snap frozen in liquid $N_2$ then 160 µm sections were cut throughout the hypothalamus at −21°C using a cryostat and the mediobasal hypothalamus (MBH) was microdissected using microdissection scissors. Tissues were mechanically homogenized in ice cold RIPA lysis buffer (50 mM Hepes [pH 7.4], 1% (v/v) Triton X-100, 1% (v/v) sodium deoxycholate, 0.1% (v/v) SDS, 150 mM NaCl, 10% (v/v) glycerol, 1.5 mM MgCl2, 1 mM EGTA, 50 mM NaF, leupeptin (5 µg/ml),

pepstatin A (1 µg/ml), 1 mM benzamadine, 2 mM phenylmethysulfonyl fluoride, 1 mM sodium vanadate) and clarified by centrifugation (100, 000 x g for 20 min at 4°C). Tissue lysates were resolved by SDS-PAGE and immunoblotted as described previously (*Loh et al., 2011*). Antibodies used are rabbit anti-phospho-IRβ-Y1162/Y1163 (p-IRβ) from ThermoScientific and rabbit anti-IR-β from Santa Cruz Biotechnology. All other antibodies were from Cell Signaling Technology.

## Quantitative PCR

RNA was extracted using TRIzol reagent or RNAzol (Sigma) and total RNA quality and quantity determined using a NanoDrop 3300 (Thermo Scientific, Wilmington, DE, USA). mRNA was reverse transcribed using a High-Capacity cDNA Reverse Transcription Kit (Applied Biosystems, Foster City, CA) and processed for quantitative real-time PCR using the TaqMan Universal PCR Master Mix and TaqMan Gene Expression Assays (Applied Biosystems, Foster City, CA). The following TaqMan gene expression assays were used: *Pomc* (Mm00435874_m1), *Npy* (Mm03048253_m1), *Agrp* (Mm00475829_g1), *Gapdh* (Mm99999915_g1), *Pck1* (Mm01247058_m1), *G6pc* (Mm00839363_m1) and *Il6* (Mm00446190_m1). Gene expression was normalized to *Gapdh*. Relative quantification was achieved using the ΔΔCT method. Reactions were performed using a BioRad CFX 384 touch (Bio-Rad, Hercules, CA).

## Acknowledgements

This work was supported by the National Health and Medical Research Council (NHMRC) of Australia (to TT, SA, MAC, and DS) and NIH to TLH (AG051459, AG052986, DK111178) and Z-YZ (RO1 CA207288); TT and MAC are NHMRC Research Fellows.

## Additional information

### Funding

| Funder | Grant reference number | Author |
| --- | --- | --- |
| National Institutes of Health | RO1 CA207288 | Zhong-Yin Zhang |
| National Health and Medical Research Council | | Michael A Cowley Sofianos Andrikopoulos David Spanswick Tony Tiganis |
| National Institutes of Health | AG051459 | Tamas L Horvath |
| National Institutes of Health | AG052986 | Tamas L Horvath |
| National Institutes of Health | DK111178 | Tamas L Horvath |

The funders had no role in study design, data collection and interpretation, or the decision to submit the work for publication.

### Author contributions

Garron T Dodd, Conceptualization, Formal analysis, Investigation, Visualization, Methodology, Writing—original draft, Writing—review and editing; Natalie J Michael, Robert S Lee-Young, Investigation, Methodology, Writing—review and editing; Salvatore P Mangiafico, Jack T Pryor, Astrid C Munder, Investigation, Methodology; Stephanie E Simonds, Resources, Methodology, Writing—review and editing; Jens Claus Brüning, Conceptualization, Resources, Writing—review and editing; Zhong-Yin Zhang, Resources, Funding acquisition, Writing—review and editing; Michael A Cowley, Sofianos Andrikopoulos, Conceptualization, Resources, Funding acquisition, Writing—review and editing; Tamas L Horvath, Conceptualization, Resources, Formal analysis, Funding acquisition, Investigation, Methodology, Writing—review and editing; David Spanswick, Conceptualization, Resources, Formal analysis, Supervision, Funding acquisition, Investigation, Methodology, Writing—review and editing; Tony Tiganis, Conceptualization, Resources, Formal analysis, Supervision, Funding acquisition, Validation, Visualization, Methodology, Writing—original draft, Project administration, Writing—review and editing

## Author ORCIDs

Garron T Dodd (iD) http://orcid.org/0000-0002-7554-4876
Natalie J Michael (iD) http://orcid.org/0000-0002-9032-0862
Tamas L Horvath (iD) http://orcid.org/0000-0002-7522-4602
Tony Tiganis (iD) http://orcid.org/0000-0002-8065-9942

## Ethics

Animal experimentation: Experiments were approved by the Monash University School of Biomedical Sciences Animal Ethics Committee (MARP2013/137).

## Decision letter and Author response

Decision letter https://doi.org/10.7554/eLife.38704.024
Author response https://doi.org/10.7554/eLife.38704.025

# Additional files

### Supplementary files

• Supplementary file 1. Statistical analysis of POMC neuronal electrophysiological responses to insulin. Statistical significance and P values (Yate's continuity corrected Chi-squared test) comparing populations of POMC neurons that are activated, inhibited or unresponsive to insulin in $Ptpn2^{fl/fl}$, POMC-TC and POMC-TC-IR mice.
DOI: https://doi.org/10.7554/eLife.38704.019

• Supplementary file 2. Statistical analysis of POMC neuronal electrophysiological responses to insulin after the inhibition of TCPTP. Statistical significance and P values (Yate's continuity corrected Chi-squared test) comparing populations of POMC neurons from chow or high fat fed (HFF) mice that are activated, inhibited or unresponsive to insulin after administration of vehicle or TCPTP inhibitor (compound 8, 20 nM).
DOI: https://doi.org/10.7554/eLife.38704.020

• Transparent reporting form
DOI: https://doi.org/10.7554/eLife.38704.021

### Data availability

All data generated or analysed during this study are included in the manuscript and supporting files.

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
