## [Decision Letter]

Thank you for submitting your article "Insulin regulates POMC neuronal plasticity to control glucose metabolism" for consideration by *eLife*. Your article has been reviewed by three peer reviewers, and the evaluation has been overseen by a Reviewing Editor and Mark McCarthy as the Senior Editor. The following individual involved in review of your submission has agreed to reveal his identity: Dong Kong (Reviewer #2).

The reviewers have discussed the reviews with one another and the Reviewing Editor has drafted this decision to help you prepare a revised submission.

This interesting manuscript presents a large amount of information regarding the role of a specific protein tyrosine phosphatase, TCPTP, in the control of insulin responsiveness in POMC neurons in relation to control of systemic insulin sensitivity and hepatic glucose metabolism. The data strongly suggest both a key role for TCPTP in the responsiveness of POMC cells to insulin and that enhanced insulin signaling depolarizes and increases the firing of in these neurons, while the opposite effect is observed in POMC neurons with high levels of TCPTP. The data also offer a welcome new perspective on the role of POMC neurons in the control of HGP and a potential resolution to controversy surrounding this issue stemming from earlier work, and the impact of changes in nutritional state on POMC cell insulin responsiveness is also novel and of interest.

The reviewers of this paper found the results to be of great interest and suitable for publication in *eLife* after some revisions. It was not possible to assemble the comments into a few key points for consideration (*eLife* normal procedure) because the three reviewers had non-overlapping concerns. Thus, all of the comments are included below. The reviewers had a number of concerns that can, in general, be addressed without further experiments. However, explanations of the many points raised are necessary and caveats to the data interpretation should be included. Additional experiments that address concerns are, of course, welcome.

One issue raised by reviewers is that the use of *Pomc*-Cre mice is problematic because Cre recombinase is activated in neurons that go to develop into AgRP (and other) neurons. Thus, some of the data are probably derived from a mixed population of arcuate neurons.

The other issue is that the mitochondria data are correlative and do not provide compelling evidence linking mitochondrial dynamics and glucose regulation. Some reviewers, including the editor, suggest either providing a causal link between mitochondrial dynamics and glucose regulation, or removing the mitochondrial experiments from this story.

The authors should define TCPTP on first usage and note the mouse gene names do not include hyphens, e.g. Il6, not Il-6.

Reviewer #1:

In this study, Dodd et al. investigate the role of TCPTP in POMC neurons upon insulin signaling and glucose metabolism control. The authors report a correlation between TCPTP expression in POMC neurons and the level of neuron responsiveness to insulin. High expression of TCPTP in obesity or fasting hampers insulin-induced signaling, POMC neuron activation and repression of HGP. In contrast, genetic inactivation of TCPTP in POMC neurons causes enhanced insulin-stimulated signaling, activation of POMC neurons and inhibition of HGP. The authors also report changes in mitochondrial dynamics and mitochondria-ER contacts in POMC neurons upon TCPTP deletion. The authors propose that physiological (fed/fast) and pathophysiological (HFD) changes in TCPTP expression in POMC neurons may influence neuronal activity and HGP via insulin signaling-mediated modulation of mitochondrial fusion and contacts with the ER.

The study uses a nice combination of mouse genetics, electrophysiology, pharmacology as well as physiological and molecular biology techniques. Generally speaking, the conclusions are supported by the data although some statements in specific parts of the manuscript should be softened. Overall, the study fulfils the highest standards in the field and seems appropriate for publication in *eLife*.

I do not have major concerns that require additional experiments but I would suggest further analysis of the electrophysiology results. In particular, the authors compare the number (population size) of POMC neurons that are activated, inhibited or unresponsive to insulin amongst a number of experimental settings. Table 1 provides statistical comparisons for some of these studies, but not all (only for KO's and double Ko's). However, the authors claim that pharmacological TCPTP inhibition "significantly altered POMC responses to insulin" but statistical evidence is not provided. The same for the HFD study. The authors should provide a complete table of comparisons amongst the diverse experimental settings.

Reviewer #2:

Insulin's effect on POMC neurons, their neuronal heterogeneity, and the related blood glucose regulation have been interesting topics in the field of brain-controlled metabolism for quite a while. However, the underlying mechanisms are still unclear. Whether POMC neurons regulate glucose homeostasis is also under debate. In the current study, Dodd et al. focused on these questions, performed a series of studies, and made multiple important and exciting findings. First, they documented heterogenous responses of POMC neurons to insulin and further demonstrated a dependence of such responses on the expression of TCPTP phosphatase in these neurons. Then, the author performed glucose clamp studies by combining with chemogenetic tools and assessed the peripheral mechanisms. They found that TCPTP in POMC neurons regulated blood glucose by mainly inhibiting hepatic glucose production. Finally, they also used multiple mouse models and discovered that feeding/fasting and HFD-feeding induced TCPTP expression in POMC neurons regulated insulin receptor signaling and mitochondrial dynamics to manipulate responses to insulin and peripheral glucose homeostasis.

Overall, this manuscript is very well prepared, the studies were properly designed, performed, and analyzed, and the conclusions were well reached. The current study has thus established an elegant system to understand the central control of glucose metabolism. I believe the manuscript will be of great interest to the readers of *eLife* and I have the following comments for the authors to address.

1) The authors performed a large number of current-clamp recordings on POMC neurons to assess their responses to insulin. However, the results of membrane potential seem problematic in the current study. For example, in the first paragraph of the subsection “TCPTP defines POMC neurons that are activated or inhibited by insulin”, POMC neurons after insulin stimulation exhibited an averaged membrane potential of -40.1+/-2.5 mV; in the next paragraph of the aforementioned subsection, excited neurons had a membrane potential as depolarized as -37.9+/-1.7 mV. How could POMC neurons be so depolarized?

2) As shown previously by Padilla, Carmody and Zeltser, et al. (Nat Med 2010, and in a more convincing way recently by Xu et al., 2018, POMC-Cre exhibited ectopic Cre activity during development, particularly in AgRP neuron. Breeding *Pomc*-Cre with floxed mice could yield a manipulation of 25% AgRP neurons as well (Xu et al.). In Figure 1G-H, since the authors used Z/EG reporter instead of *Pomc*-eGFP transgenic mice to indicate POMC neurons, AgRP neurons were expected to be patched on as well. Since AgRP neurons are known to be inhibited by insulin, the ratio for the insulin-inhibited POMC neurons (12/33, 36.4%) may need to be adjusted.

3) Subsection “The activation or inhibition of POMC neurons regulates hepatic glucose metabolism”, the authors utilized inhibitory DREADDs to inhibit POMC neurons and included convincing expression data of hM4Di in Figure 2—figure supplement 1. However, a direct test of CNO's activity on POMC neurons with current-clamp recording shall also be included since hM4Di works through the activation of GIRK channels and not all neurons express them. In addition, CNO was recently reported to activate other endogenous receptors as well (Gomez et al., Science 2017). CNO's effects on non-DREADD-expressing animals shall also be included as a control.

4) In the third paragraph of the Discussion, the authors discussed about the potential mechanisms with regards to synaptic regulation. Deletion of TCPTP in POMC neurons clearly suggested a post-synaptic mechanism but the authors believed in an existence of pre-synaptic regulation. This is confusing. Do the authors think of the involvement of a retrograde mechanism or different action sites of TCPTP from POMC neurons?

Reviewer #3:

This interesting manuscript presents a large amount of information regarding the role of a specific protein tyrosine phosphatase, TCPTP, in the control of insulin responsiveness in POMC neurons in relation to control of systemic insulin sensitivity and hepatic glucose metabolism. The data strongly suggest both a key role for TCPTP in the responsiveness of POMC cells to insulin and that enhanced insulin signaling depolarizes and increases the firing of in these neurons, while the opposite effect is observed in POMC neurons with high levels of TCPTP. The data also offer a welcome new perspective on the role of POMC neurons in the control of HGP and a potential resolution to controversy surrounding this issue stemming from earlier work, and the impact of changes in nutritional state on POMC cell insulin responsiveness is also novel and of interest. These considerable strengths are offset to some extent by the following concerns.

1) A key unanswered question is how increased insulin receptor signal transduction associated with reduced TCPTP is tied to a reversal of the effect of insulin on POMC neuron membrane potential and firing rate (from inhibitory to stimulatory). The authors state that this "phenotype switch" in the insulin response reflects direct (rather than an indirect) effects of insulin, but how increased IR-PI3K-Akt signal transduction in these cells might achieve this effect is unresolved. In an effort to shed light on this key question, the authors investigate whether changes in POMC cell mitochondrial dynamics underlie this phenotype switch. Although changes of insulin responsiveness appear to be coupled to morphological and genomic alterations affecting mitochondria, the data fail to establish a connection between these mitochondrial responses and changes of membrane potential and firing rate. Consequently, the data on mitochondrial dynamics seem somewhat disconnected from the rest of the paper, and might be more appropriate for a separate publication.

2) The authors finding of a robust effect of POMC-specific TCPTP deletion to enhance insulin suppression of HGP begs the question as to what other responses to insulin might also be affected. Given that ICV insulin has previously been reported to reduce food intake, body weight and body adiposity, and that brain-specific IR deletion promotes positive energy balance and weight gain, one wonders whether the effect of ICV insulin on food intake, body weight and body adiposity are also increased. Given the marked improvement of whole-body insulin sensitivity attributed to POMC-specific TCPTP deletion, one also wonders why the basal glucose and insulin levels were not reduced in these animals.

3) A related question is whether reduced body adiposity results from POMC-specific TCPTP deletion, and if so, did this contribute to enhanced insulin-induced suppression of HGP reported by the authors? Conversely, if POMC-specific TCPTP deletion has no effect on energy balance or body adiposity, this seems surprising given the expected increase of POMC neuron firing.

4) The question of how pronounced changes of POMC expression of TCPTP are associated with feeding, fasting and DIO was not addressed. Doing so would considerably strengthen the manuscript and seems of much greater relevance than the large amount of mitochondrial data that were included.

5) During embryogenesis, the POMC promoter is expressed in multiple arcuate nucleus neuronal subtypes including AgRP neurons. Since POMC-Cre was used to delete TCPTP, one expects the gene to be deleted across multiple arcuate nucleus neuronal cell types. Some effort to address this concern is warranted.

6) The authors state that diet-induced obesity reduces POMC neuron insulin responsiveness via a mechanism involving induction of TCPTP. Is this effect of DIO causally linked to associated responses such as inflammation, reactive gliosis, ER stress, etc.? Discussion of this question seems warranted.

7) The authors report (Discussion, last paragraph) that TCPTP deletion has no detectable effect on glucose homeostasis in DIO mice, which they attribute to the fact that other mechanisms can account for impaired POMC insulin responsiveness in this setting. But they stop short of investigating whether TCPTP deletion enhances insulin responsiveness to POMC neurons in this setting. This question should be addressed since, if POMC neuron insulin responsiveness is in fact restored and yet there is no whole-body phenotype, the physiological relevance of POMC neuron insulin responsiveness would be called into question.

8) What precedent exists for a change in hormone responsiveness reversing its effect on the firing of a distinct neuronal subset? If such a response has been reported previously, was it referred to as "neural plasticity"? Certainly, this is not the type of response that comes to mind in association with that term. In this context, the term "neural plasticity" seems potentially misleading and for this reason, it is recommended that it be removed from the manuscript title.

9) The effect of IP insulin to induce c-Fos in the PVH (Figure 1M) could have resulted from hypoglycemia, which was presumably well established by 90 min following the dose of insulin that was given. The effect of neuroglucopenia to activate PVH neurons is well established (e.g., Briski, Neuroreport, 1998; Briski, Brain Res Bulletin, 2000; Evans, AJP, 2001), and it is conceivable that changes of POMC cell insulin responsiveness affected the degree of insulin-induced hypoglycemia.

10) When insulin is given ICV and HGP or other highly insulin-responsive endpoints are measured, it's important to verify that none of the centrally-administered insulin leaked into the periphery to have direct effects on the liver.

11) Relevant to this point is the use of a study design in which ICV insulin (0.1 mU) or vehicle was given as 5 separate injections over 5 hours. It is difficult to imagine that 5 ICV injections over 5 hours would be well-tolerated, which in turn raises the possibility that the authors were inadvertently measuring the impact of insulin on a stress response, which might explain the basal hyperglycemia reported in some studies. Beyond this, insulin in CSF has a relatively long half-life, so it's not clear why a repeated injection protocol over 5 hours was required. Were the animals anesthetized during this period? I could not find the information in the Materials and methods section.

12) The basal glucose level (time=0 min) of both groups in the pyruvate tolerance test shown in Figure 2C is ~12 mM, which is in the diabetic range. By comparison, basal glucose levels of 7 mM are reported for the same test in Figure 2J, and are uniformly <10 mM in all other such studies (Figures 4, 5 and 6). This discrepancy is all the more notable in light of the remarkably small variances (SEMs) around the glucose data, which is inconsistent with known biological variability inherent in the response.

13) The Discussion is longer than it need be, and tends to reiterate points made in the Results section. It could be cut substantially.

---

## [Author Response]

[…] The reviewers of this paper found the results to be of great interest and suitable for publication in eLife after some revisions. It was not possible to assemble the comments into a few key points for consideration (eLife normal procedure) because the three reviewers had non-overlapping concerns. Thus, all of the comments are included below. The reviewers had a number of concerns that can, in general, be addressed without further experiments. However, explanations of the many points raised are necessary and caveats to the data interpretation should be included. Additional experiments that address concerns are, of course, welcome.One issue raised by reviewers is that the use of Pomc-Cre mice is problematic because Cre recombinase is activated in neurons that go to develop into AgRP (and other) neurons. Thus, some of the data are probably derived from a mixed population of arcuate neurons.

We have now discussed this caveat in relation to the glucose metabolism studies in our POMC TCPTP knockout mice in the revised Discussion. Importantly, we make the specific point in the results that *Pomc*-eGFP faithfully marks POMC neurons (Padilla et al., 2010) and that *Pomc*-Cre in adult mice deletes exclusively in POMC neurons (Xu et al., 2018). Accordingly our studies defining the existence of POMC neurons that are activated or inhibited by insulin, the increased proportion of activated POMC neurons after TCPTP inhibition, the decreased proportion of POMC neurons in obesity that is corrected by TCPTP inhibition and the capacity of activating and inhibitory DREADDs to alter glucose metabolism are not compromised by the off target effects of the *Pomc*-Cre transgene. Collectively, our studies are entirely consistent with alterations in insulin-induced POMC neuronal excitability regulating hepatic glucose metabolism, but we have acknowledged that some of the effects on glucose metabolism in POMC-TC mice might be due to deletion in AgRP/NPY neurons.

The other issue is that the mitochondria data are correlative and do not provide compelling evidence linking mitochondrial dynamics and glucose regulation. Some reviewers, including the editor, suggest either providing a causal link between mitochondrial dynamics and glucose regulation, or removing the mitochondrial experiments from this story.

We agree with the editor and reviewers. Our goal was to define the mechanisms by which TCPTP could functionally modulate POMC neuronal excitability but we concede that our findings in this regard although exciting are correlative. We have revised the manuscript omitting the mitochondrial dynamics data. However, we have discussed the possibility that mitochondrial dynamics may mechanistically coordinate POMC neuronal excitability in the Discussion.

The authors should define TCPTP on first usage and note the mouse gene names do not include hyphens, e.g. Il6, not Il-6.

We have changed all gene names not to include hyphens and have defined TCPTP on fits usage in the Introduction.

Reviewer #1:[…] I do not have major concerns that require additional experiments but I would suggest further analysis of the electrophysiology results. In particular, the authors compare the number (population size) of POMC neurons that are activated, inhibited or unresponsive to insulin amongst a number of experimental settings. Table 1 provides statistical comparisons for some of these studies, but not all (only for KO's and double Ko's). However, the authors claim that pharmacological TCPTP inhibition "significantly altered POMC responses to insulin" but statistical evidence is not provided. The same for the HFD study. The authors should provide a complete table of comparisons amongst the diverse experimental settings.

We have now included an additional table (Figure 1—figure supplement 2) detailing the statistical differences between POMC neurons that are excited, inhibited or non-responsive in response to insulin following either treatment with TCPTP inhibitor (Figure 1l) or high fat feeding (Figure 1K).

Reviewer #2:[…] 1) The authors performed a large number of current-clamp recordings on POMC neurons to assess their responses to insulin. However, the results of membrane potential seem problematic in the current study. For example, in the first paragraph of the subsection “TCPTP defines POMC neurons that are activated or inhibited by insulin”, POMC neurons after insulin stimulation exhibited an averaged membrane potential of -40.1+/-2.5 mV; in the next paragraph of the aforementioned subsection, excited neurons had a membrane potential as depolarized as -37.9+/-1.7 mV. How could POMC neurons be so depolarized?

The data are presented as the absolute read-out at the time of experiments and do not account for the liquid junction potential, which for our recording conditions amounts to 10/11mV. Thus, the values are depolarised. To acknowledge this, we have now incorporated the following sentence into the electrophysiological recordings section in the supplemental experimental procedures section, “Resting membrane potential is expressed as the read-out from the amplifier and was not corrected for the liquid junction potential offset (11 mV).”

2) As shown previously by Padilla, Carmody and Zeltser, et al. (Nat Med 2010, and in a more convincing way recently by Xu et al., 2018, POMC-Cre exhibited ectopic Cre activity during development, particularly in AgRP neuron. Breeding Pomc-Cre with floxed mice could yield a manipulation of 25% AgRP neurons as well (Xu et al.). In Figure 1G-H, since the authors used Z/EG reporter instead of Pomc-eGFP transgenic mice to indicate POMC neurons, AgRP neurons were expected to be patched on as well. Since AgRP neurons are known to be inhibited by insulin, the ratio for the insulin-inhibited POMC neurons (12/33, 36.4%) may need to be adjusted.

See response to Editors. The reviewer is correct. Although we have discussed the caveat associated with the off-target effects of the *Pomc*-Cre transgene in POMC-TC mice on glucose metabolism in the Discussion and included a specific statement in the results highlighting the fact that we cannot exclude a proportion of inhibited neurons being AgRP neurons, we are reluctant to retrospectively adjust the number of neurons inhibited by insulin in POMC-TC;ZE/G mice without staining for NPY/AgRP neurons. Irrespective this would only under represent the number of POMC neurons being activated by insulin. Moreover to complement our findings we have also assessed POMC neuronal excitation/inhibition in *Pomc*-eGFP mice treated with the TCPTP inhibitor where we similarly show that TCPTP inhibition is accompanied by an increase in POMC neurons being activated by insulin.

3) Subsection “The activation or inhibition of POMC neurons regulates hepatic glucose metabolism”, the authors utilized inhibitory DREADDs to inhibit POMC neurons and included convincing expression data of hM4Di in Figure 2—figure supplement 1. However, a direct test of CNO's activity on POMC neurons with current-clamp recording shall also be included since hM4Di works through the activation of GIRK channels and not all neurons express them. In addition, CNO was recently reported to activate other endogenous receptors as well (Gomez et al., Science 2017). CNO's effects on non-DREADD-expressing animals shall also be included as a control.

hM4Di has been previously validated in POMC neurons (Koch et al., 2015, and Atasoy et al., 2012). These previous studies showed conclusively that in response to CNO administration, hM4Di attenuates c-Fos expression (a surrogate marker of neuronal activity) in ARC POMC neurons and reduces POMC neuronal firing rate as determined by current-clamp recordings. We have also conducted further electrophysiological experiments showing POMC neurons that do not express hM4Di, fail to respond to CNO. This indicates that CNO is not having any off-target effects and is not activating non-specific endogenous receptors on POMC neurons (Author response image 1).

**Author response image 1. respfig1:** CNO has no effect of firing frequency and membrane potential in non-DREADD-expressing POMC neurons. Representative traces of whole-cell patch clamp recordings of individual hypothalamic non-DREADD expressing GFP positive POMC neurons in response to CNO (10 µM) in *Pomc*-Cre;*Pomc*-eGFP mice. CNO had no effects in all (11 cells, n=4) non-DREADD POMC neurons recorded.

To address the reviewer’s concern that the CNO metabolite, clozapine, may be having off target effects on glucose metabolism, we conducted GTTs, ITTs and PTTs in non-DREADD expressing *Pomc*-Cre mice following administration of vehicle or CNO (0.3mg/kg, i.p., 2 injections over 6h, Figure 2—figure supplement 1C-E). We found that CNO had no significant effect on whole body glucose metabolism (Figure 2—figure supplement 1C-E).

These findings are consistent with recent a recent study [Rossi M et al., (2018) J Clin Invest] that showed that a dose of 10 mg/kg (i.p.) CNO has no effect on fed/fasted blood glucose levels or glucose metabolism. The dose of CNO used in our study (0.3mg/kg injected twice over 6h) is ~33x lower than that described by Rossi et al. Furthermore, our results define a biphasic modulation of glucose metabolism by activating or inhibitory DREADDs in POMC neurons. It is highly unlikely that CNO alone elicits a biphasic regulation of glucose metabolism.

4) In the third paragraph of the Discussion, the authors discussed about the potential mechanisms with regards to synaptic regulation. Deletion of TCPTP in POMC neurons clearly suggested a post-synaptic mechanism but the authors believed in an existence of pre-synaptic regulation. This is confusing. Do the authors think of the involvement of a retrograde mechanism or different action sites of TCPTP from POMC neurons?

This was merely speculation as to mechanisms involved. Our EM analyses indicated that there was no overt change in relative excitatory and inhibitory inputs into POMC perikarya. However, beyond this, we did not specifically investigate presynaptic mechanisms but instead detailed postsynaptic changes in excitability. Exploring the specific mechanisms involved extends beyond the scope of this study.

Reviewer #3:[…] 1) A key unanswered question is how increased insulin receptor signal transduction associated with reduced TCPTP is tied to a reversal of the effect of insulin on POMC neuron membrane potential and firing rate (from inhibitory to stimulatory). The authors state that this "phenotype switch" in the insulin response reflects direct (rather than an indirect) effects of insulin, but how increased IR-PI3K-Akt signal transduction in these cells might achieve this effect is unresolved. In an effort to shed light on this key question, the authors investigate whether changes in POMC cell mitochondrial dynamics underlie this phenotype switch. Although changes of insulin responsiveness appear to be coupled to morphological and genomic alterations affecting mitochondria, the data fail to establish a connection between these mitochondrial responses and changes of membrane potential and firing rate. Consequently, the data on mitochondrial dynamics seem somewhat disconnected from the rest of the paper, and might be more appropriate for a separate publication.

We agree with the reviewer and have omitted the mitochondrial dynamics data.

2) The authors finding of a robust effect of POMC-specific TCPTP deletion to enhance insulin suppression of HGP begs the question as to what other responses to insulin might also be affected. Given that ICV insulin has previously been reported to reduce food intake, body weight and body adiposity, and that brain-specific IR deletion promotes positive energy balance and weight gain, one wonders whether the effect of ICV insulin on food intake, body weight and body adiposity are also increased. Given the marked improvement of whole-body insulin sensitivity attributed to POMC-specific TCPTP deletion, one also wonders why the basal glucose and insulin levels were not reduced in these animals.

We have previously published that TCPTP deletion in POMC neurons has no effect on overnight food intake, adiposity or body weight [Dodd et al., 2015]. Nonetheless to address the reviewer’s concern, we have assessed the effects of ICV insulin (0.1mU/animal, 5 injections over 5 h as used in Figure 3, Figure 5G-H, Figure 6G-J, Figure 7A-D) on food intake, body weight and adiposity in 8-week-old male *Ptpn2^fl/fl^*, POMC-TC and POMC-TC-IR mice (Figure 6—figure supplement 3A-D). Although the administration of insulin ICV had marked effects on glucose metabolism in POMC-TC mice, this was not accompanied by a reduction in body weight, food intake or whole-body adiposity at this dose and timeframe (Figure 6—figure supplement 3A-D).

As described in the manuscript, we noted a decrease in fasted plasma glucose and plasma insulin levels in POMC-TC mice (Figure 4A-B). We now also show that fasted blood glucose and plasma insulin levels are also reduced in high fat fed obese mice (Figure 7).

3) A related question is whether reduced body adiposity results from POMC-specific TCPTP deletion, and if so, did this contribute to enhanced insulin-induced suppression of HGP reported by the authors? Conversely, if POMC-specific TCPTP deletion has no effect on energy balance or body adiposity, this seems surprising given the expected increase of POMC neuron firing.

We have previously published that TCPTP deletion in POMC neurons has no effects on adiposity or bodyweight [Dodd et al., 2017) Cell]. TCPTP deletion in POMC neurons is only accompanied by the promotion of energy expenditure and decreased adiposity when accompanied by the concomitant enhancement of leptin signalling. Exploring the basis for this extends beyond the scope of this study. No differences in body weight or adiposity were evident in POMC-TC mice used in this study, precluding any changes in endogenous glucose production being related to this.

4) The question of how pronounced changes of POMC expression of TCPTP are associated with feeding, fasting and DIO was not addressed. Doing so would considerably strengthen the manuscript and seems of much greater relevance than the large amount of mitochondrial data that were included.

We have previously published that changes in POMC TCPTP expression are associated with feeding, fasting and diet-induced obesity [Dodd et al., 2017]; we have referenced this in the manuscript. We have shown that heightened glucocorticoid (corticosterone) levels in the fasted state drive the expression of TCPTP in hypothalamic neurons to suppress insulin signalling. By contrast feeding is associated with decreased glucocorticoid levels and concomitant *Ptpn2* gene expression and the degradation of TCPTP protein. In the obese state we have shown that the feeding induced repression of TCPTP is abrogated so that TCPTP levels are always high emulating the fasted state; the precise reason for this is unclear, but probably occurs as a consequence of the heightened glucocorticoid and leptin levels [Loh et al., 2011] in obesity, which we have also shown drive hypothalamic TCPTP expression. We have discussed this in the revised manuscript

5) During embryogenesis, the POMC promoter is expressed in multiple arcuate nucleus neuronal subtypes including AgRP neurons. Since POMC-Cre was used to delete TCPTP, one expects the gene to be deleted across multiple arcuate nucleus neuronal cell types. Some effort to address this concern is warranted.

See responses to editors and reviewer 2 (point 2).

6) The authors state that diet-induced obesity reduces POMC neuron insulin responsiveness via a mechanism involving induction of TCPTP. Is this effect of DIO causally linked to associated responses such as inflammation, reactive gliosis, ER stress, etc.? Discussion of this question seems warranted.

See point 4. We have discussed this in the revised manuscript.

7) The authors report (Discussion, last paragraph) that TCPTP deletion has no detectable effect on glucose homeostasis in DIO mice, which they attribute to the fact that other mechanisms can account for impaired POMC insulin responsiveness in this setting. But they stop short of investigating whether TCPTP deletion enhances insulin responsiveness to POMC neurons in this setting. This question should be addressed since, if POMC neuron insulin responsiveness is in fact restored and yet there is no whole-body phenotype, the physiological relevance of POMC neuron insulin responsiveness would be called into question.

Our studies indicated that TCPTP deficiency in POMC neurons enhanced the ICV insulin induced repression of HGP as assessed in PTTs and measuring hepatic gluconeogenic gene expression. We also found that although fasted blood glucose levels (and now in revised manuscript fasted plasma insulin levels) were reduced, ITT and GTT responses were not altered. ITTs and GTTs are crude measure of glucose homeostasis and primarily measure glucose clearance mediated by skeletal muscle. Given this and the reviewer’s concern re physiological relevance we subjected 12-week high fat fed obese mice to hyperinsulinemic euglycemic clamps. We found that glucose infusion rates were significantly increased in POMC-TC mice (Figure 7G, Figure 7—figure supplement 1E), consistent with enhanced systemic insulin sensitivity. This was attributable exclusively to the suppression of HGP and reduced gluconeogenic (*Pck1* and *G6pc*) gene expression (Figure 7J). Taken together our findings are consistent with elevated TCPTP in obesity perturbing POMC insulin responses to drive HGP and fasting hyperglycemia in obesity.

8) What precedent exists for a change in hormone responsiveness reversing its effect on the firing of a distinct neuronal subset? If such a response has been reported previously, was it referred to as "neural plasticity"? Certainly, this is not the type of response that comes to mind in association with that term. In this context, the term "neural plasticity" seems potentially misleading and for this reason, it is recommended that it be removed from the manuscript title.

The term neural plasticity is generally attributed to changes in synaptic efficacy, for example long-term potentiation or depression (LTP and LTD, respectively). However, the term plasticity itself by definition refers to changes in neuronal function, be it learning and memory, development whatever. This is clearly as we have seen in our study in relation to hormone responsiveness and therefore we consider the term entirely appropriate. Clearly these neurons do show plasticity in their responsiveness to hormones, which leads to changes in physiological function. Whilst we understand the referee’s point of view in that tradition has restricted the use of the term to phenomena such as LTP and LTD etc. we believe our use of the term to relate to functional plasticity of hormone responsiveness under a range of physiological conditions entirely appropriate. With regards to precedents for this, hormone responsiveness (noradrenaline) has previously been shown to reverse (excitation to inhibition) in orexin neurons following a short period of sleep deprivation (Grivel et al., 2005). In a metabolic context, glucose sensing responses in ARC neurons change depending on metabolic status in an energy-status-dependent manner (van den Top et al., 2017). These studies show a functional plasticity in these neural populations and set a precedence for such changes in response to hormones or nutrients in specific neuronal populations. We also believe that we are leading the way in driving awareness of the functional, energy-status-dependent plasticity of energy-sensing neural networks. We therefore respectfully consider our use of this term is entirely appropriate.

9) The effect of IP insulin to induce c-Fos in the PVH (Figure 1M) could have resulted from hypoglycemia, which was presumably well established by 90 min following the dose of insulin that was given. The effect of neuroglucopenia to activate PVH neurons is well established (e.g., Briski, Neuroreport, 1998; Briski, Brain Res Bulletin, 2000; Evans, AJP, 2001), and it is conceivable that changes of POMC cell insulin responsiveness affected the degree of insulin-induced hypoglycemia.

To address the reviewers concerns we administered a dose of 0.85 mU/g insulin (as described in Figure 1M) and measured blood glucose levels 90 min post injection in *Ptpn2^fl/fl^* and POMC-TC mice (Figure 1—figure supplement 4A). We found that the insulin-induced hypoglycemia was similar in *Ptpn2^fl/fl^* and POMC-TC mice. Therefore differences in hypoglycemia could not account for the enhanced PVH c-Fos immunoreactivity seen in POMC-TC mice (Figure 1M).

10) When insulin is given ICV and HGP or other highly insulin-responsive endpoints are measured, it's important to verify that none of the centrally-administered insulin leaked into the periphery to have direct effects on the liver.

We thank the reviewer for raising this important point. To address this concern, we fasted 8-week-old male C57BL/6J mice and administered ICV saline or insulin (0.1 mU/animal, 5 injections over 5 h, as indicated in Figure 3—figure supplement 1A-B as in Figure 3A-D, Figure 5A-H, Figure 7G-J, Figure 7B and Figure 3—figure supplement 1C) and plasma insulin levels were determined at 0, 15, 30, 60 and 120 min post-injection. Plasma insulin levels were determined using an insulin ELISA (Monash Antibody Technologies Facility) capable of detecting both mouse and human insulin.

We found that no significant difference in plasma insulin level between vehicle and ICV insulin treated mice (Figure 3—figure supplement 1A-B), indicated that the effects on liver glucose metabolism that we describe inFigure 3A-D, Figure 5A-H, Figure 7G-J, Figure 7B and Figure 3—figure supplement 1Care unlikely to result from insulin leaking into the periphery.

11) Relevant to this point is the use of a study design in which ICV insulin (0.1 mU) or vehicle was given as 5 separate injections over 5 hours. It is difficult to imagine that 5 ICV injections over 5 hours would be well-tolerated, which in turn raises the possibility that the authors were inadvertently measuring the impact of insulin on a stress response, which might explain the basal hyperglycemia reported in some studies. Beyond this, insulin in CSF has a relatively long half-life, so it's not clear why a repeated injection protocol over 5 hours was required. Were the animals anesthetized during this period? I could not find the information in the Materials and methods section.

Whilst it is likely that some level of stress will be caused by 5x separate ICV injections over 5 hours, we minimized handling stress by handling the mice 2 weeks prior to the experiment and conducting dummy injections 2 days prior to the experiment. All treatment groups received repeated ICV injection of either vehicle or insulin and were thus subjected to the same amount of handling stress. We conclude that handling stress is not responsible for the effects of TCPTP deletion in POMC neurons on glucose metabolism described in Figure 3A-D, Figure 5A-H, Figure 7G-J, Figure 7B and Figure 3—figure supplement 1C. Furthermore, the enhanced insulin-POMC-mediated repression of hepatic glucose production (as assessed by the ability of ICV insulin to repress pyruvate-induced glucose excursions and hepatic gluconeogenic gene expression) was corrected in POMC-TC-IR mice (Figure 6G-I) reaffirming the importance of insulin signalling.

Mice were not anaesthetized during injections as anesthesia has confounding actions on glucose homeostasis and 5 repeated injections over 5 h of a low dose of insulin was preferred to a single high dose of insulin which could potentially result in neuroglucopenia or result in a confounding amount of insulin “spill over’ into the peripheral circulation.

12) The basal glucose level (time=0 min) of both groups in the pyruvate tolerance test shown in Figure 2C is ~12 mM, which is in the diabetic range. By comparison, basal glucose levels of 7 mM are reported for the same test in Figure 2J, and are uniformly <10 mM in all other such studies (Figures 4, 5 and 6). This discrepancy is all the more notable in light of the remarkably small variances (SEMs) around the glucose data, which is inconsistent with known biological variability inherent in the response.

We would like to make clear that the discrepancy between the basal glucose levels shown in Figure 2C and Figure 2J is due to the fact that the pyruvate tolerance tests (PTT) depicted in Figure 2C were conducted in the fed state (where blood glucose levels would be elevated), whereas the PTTs depicted in Figure 2J are were conducted in the fasted state (where glucose levels would be comparatively lower).

Figure 2 is looking at the biphasic actions of the pharmacogenetic modulation of POMC neurons on hepatic glucose production (HGP) using a PTT as a surrogate measure of gluconeogenesis. We reasoned that POMC neuronal inhibition would promote hepatic glucose production whereas POMC excitation would attenuate HGP. In order to tease apart these differences, mice were placed in opposing nutritional states, fed verse fasted, (Figures2A, B, I) which is essential to overcome any confounding aspects of endogenous HGP.

13) The Discussion is longer than it need be, and tends to reiterate points made in the Results section. It could be cut substantially.

The Discussion has been shortened.